# Global ocean surface heat fluxes derived from the maximum entropy production framework accounting for ocean heat storage and Bowen ratio adjustments

Yong Yang[1], Huaiwei Sun[1,2,3,4*], Jingfeng Wang[5], Wenxin Zhang[6], Gang Zhao[7], Weiguang Wang[8], Lei Cheng[9], Lu Chen[1], Hui Qin[1], Zhanzhang Cai[6]

[1]School of Civil and Hydraulic Engineering, Huazhong University of Science and Technology, Wuhan 430074, China

[2]Hubei Key Laboratory of Digital River Basin Science and Technology, Huazhong University of Science and Technology, Wuhan 430074, China

[3]Institute of Water Resources and Hydropower, Huazhong University of Science and Technology, Wuhan 430074, China

[4]College of Water Conservancy and Architectural Engineering, Shihezi University, Shihezi 832003, China

[5.] School of Civil and Environmental Engineering, Georgia Institute of Technology, Atlanta 30318, USA

[6.] Department of Physical Geography and Ecosystem Science, Lund University, Lund 22100, Sweden

[7.] Key Laboratory of Water Cycle and Related Land Surface Processes, Institute of Geographic Sciences and Natural Resources Research, Chinese Academy of Sciences, Beijing 100101, China

[8.] College of Hydrology and Water Resources, Hohai University, Nanjing 210098, China

[9.] State Key Laboratory of Water Resources and Hydropower Engineering Science, Wuhan University, Wuhan 430074, China

*Correspondence to*: Huaiwei Sun (hsun@hust.edu.cn) and Wenxin Zhang (wenxin.zhang@nateko.lu.se)

**Abstract.** Ocean evaporation, represented by latent heat flux (*LE*), plays a crucial role in global precipitation patterns, water cycle dynamics, and energy exchange processes. However, existing bulk methods for quantifying ocean evaporation are associated with considerable uncertainties. The Maximum Entropy Production (MEP) theory provides a novel framework for estimating surface heat fluxes, but its application over ocean surfaces remains largely unvalidated. Given the substantial heat storage capacity of the deep ocean, which can create temporal mismatches between variations in heat fluxes and radiation, it is crucial to account for heat storage when estimating heat fluxes. This study derived global ocean heat fluxes using the MEP theory, incorporating the effects of heat storage and adjustments to the Bowen ratio (the ratio of sensible heat to latent heat). We utilized multi-source data from seven auxiliary turbulent flux datasets and 129 globally distributed buoy stations to refine and validate the MEP model. The model was first evaluated using observed data from buoy stations, and the Bowen ratio formula that most effectively enhances the model performance was identified. By incorporating the heat storage effect and adjusting the Bowen ratio within the MEP model, the accuracy of the estimated heat fluxes was significantly improved, achieving an $R^2$ of 0.99 (regression

slope: 0.97) and a root mean squared error (RMSE) of 4.7 W·m$^{-2}$ compared to observations. The improved
MEP method successfully addressed the underestimation of *LE* and the overestimation of sensible heat by
the original model, providing new global estimates of *LE* at 93 W·m$^{-2}$ and sensible heat at 12 W·m$^{-2}$ for the
annual average from 1988 to 2017. Compared to the 129 buoy stations, the MEP-derived global *LE* dataset
achieved the highest accuracy, with a mean error (ME) of 1.3 W·m$^{-2}$, an RMSE of 15.9 W·m$^{-2}$, and a Kling-
Gupta Efficiency (KGE) of 0.89, outperforming four major long-term global heat flux datasets, including J-
OFURO3, ERA5, MERRA2, and OAFlux. Analysis of long-term trends revealed a significant increase in
global ocean evaporation from 1988 to 2010 at a rate of 3.58 mm·yr$^{-1}$, followed by a decline at -2.18 mm·yr$^{-}$
$^{1}$ from 2010 to 2017. This dataset provides a new benchmark for the ocean surface energy budget and is
expected to be a valuable resource for studies on global ocean warming, sea surface-atmosphere energy
exchange, the water cycle, and climate change. The 0.25° monthly global ocean heat flux dataset based on
the Maximum Entropy Production method (GOHF-MEP) for 1988–2017, is publicly accessible at
https://doi.org/10.6084/m9.figshare.26861767.v2 (Yang et al., 2024; last accessed: 28 August 2024).
**1. Introduction**
The ocean system plays a pivotal role in regulating the global climate by receiving and redistributing heat
and freshwater, thereby influencing Earth's energy balance and the dynamics of the water cycle (Li et al.,
2023; Von Schuckmann et al., 2023; Marti et al., 2022; Johnson et al., 2020). A key component in this system
is ocean evaporation, which accounts for approximately 86% of atmospheric water vapor, being the primary
driver of the global hydrological cycle (Yu, 2011). As climate change warms the ocean, evaporation rates are
expected to rise, potentially intensifying the global hydrological cycle (Masson-Delmotte et al., 2021). This
intensification could alter precipitation patterns, affecting regional water availability and freshwater
ecosystems (Konapala et al., 2020; Roderick et al., 2014). Therefore, precise estimation of ocean evaporation
is critical to understand and quantify the global energy and water budget (Iwasaki et al., 2014).
Existing methods for calculating surface latent heat (*LE*) and sensible heat flux (*H*) rely on bulk transfer
formulations, which require extensive input variables and parameterizations, such as temperature gradients,
humidity gradients, wind speed, and transfer coefficients (Fairall et al., 1996; Andreas et al., 2008). Although
widely used, these bulk methods encounter significant limitations primarily due to challenges in accurately
parameterizing and empirically deriving transfer coefficients (Zeng et al., 1998; Robertson et al., 2020).
These methods heavily depend on assumptions regarding atmospheric stability and boundary layer dynamics,
which may not consistently apply across diverse and complex environmental conditions (Fairall et al., 2003;
Andreas et al., 2013). Furthermore, uncertainties in estimating turbulent transfer coefficients can lead to
substantial errors in the estimation of latent heat flux. The high demands for parameterization and challenges
in data acquisition contribute to considerable uncertainties when implementing bulk methods for calculations.
While numerous energy balance-based algorithms have been developed to estimate global terrestrial
evapotranspiration (Wang et al., 2012; Yang et al., 2023), their application to ocean surface heat flux
estimation remains limited. Therefore, proposing an innovative method for estimating ocean surface heat flux
based on surface energy balance could yield significant theoretical and practical implications. Such an
approach could serve as a valuable complement to existing bulk methods and their associated datasets,
providing a fresh methodological perspective for quantifying ocean heat flux. This advancement would not
only enhance our ability to estimate ocean energy fluxes with greater accuracy but also deepen our
understanding of their role in the global energy and water cycles.
The Maximum Entropy Production (MEP) model, an energy-balance-based approach, has recently
emerged as a novel method for simulating surface heat fluxes. Developed from Bayesian probability theory
and information theory, the MEP prioritizes the most probable partitioning of radiation fluxes (Wang & Bras,
2011). The MEP model has been rigorously validated across diverse surface types and varying degrees of
surface wetness (Wang et al., 2014; Huang et al., 2017; Yang et al., 2022; Sun et al., 2022; Sun et al., 2023).
Notably, the MEP model requires fewer input variables—net radiation, surface temperature, and specific
humidity—yet provides accurate estimates of $LE$, $H$, and ground heat fluxes simultaneously. Unlike bulk
methods (Fairall et al., 2003), which rely on wind speed, temperature gradient, and humidity gradient, the
MEP model satisfies the energy balance constraint without these dependencies. This characteristic enhances
its applicability and robustness across diverse environmental conditions. However, the previous application
of the MEP model over ocean surfaces has revealed significant limitations, including notable
underestimations of latent heat and overestimations of $H$ (Huang et al., 2017). The global multi-year averaged
$LE$ estimated by the MEP model indicated a value around 58 W·m$^{-2}$, much lower than the range of 92~109
W·m$^{-2}$ reported by remote sensing or reanalysis-based products. Conversely, MEP estimated an averaged $H$
of approximately 28 W·m$^{-2}$, substantially higher the 6-18 W·m$^{-2}$ range reported in other studies. These
discrepancies highlight substantial uncertainties in applying the MEP model to oceanic energy partitioning,
highlighting the urgent need for further refinement and rigorous validation. These substantial errors in MEP-
estimated oceanic fluxes may be attributed to the lack of consideration of heat storage effects. The significant
impact of heat storage in deep ocean water can introduce substantial bias in estimating seasonal evaporation
rates when using the Penman combination-based method (McMahon et al., 2013; Bai & Wang et al., 2023).
For instance, deep-water bodies typically store heat during the spring and release it in the fall, which can lead
to overestimation of evaporation in the summer and underestimation in the fall if changes in heat storage are
not accounted for (Zhao & Gao, 2019; Morton, 1994). Therefore, when estimating heat fluxes using the
Bowen ratio ($B_o$, defined as the ratio of $H$ to $LE$) energy budget-based method (including the MEP method),
it is essential to incorporate heat storage effects to ensure accurate partitioning of available energy.
$B_o$ is crucial for understanding the global ocean energy partitioning process (Hicks & Hess, 1977). In
the context of the energy balance-based MEP model, the significant overestimation of $B_o$ suggests that
focusing on this ratio can enhance our understanding of energy partitioning dynamics (Andreas et al., 1996).
Studies have highlighted that the actual Bowen ratio over ocean surfaces ($B_{oa}$) often diverges from the
equilibrium Bowen ratio ($B_o^*$) observed under ideal conditions where the air is saturated with water vapor.
The $B_{oa}$ may deviate significantly from $B_o^*$ under non-equilibrium conditions, which are typical in most
environments (Jo et al., 2002; Andreas et al., 2013), posing challenges in establishing a robust relationship
between $B_{oa}$ and $B_o^*$ (Liu & Yang, 2021). Therefore, developing an accurate $B_o^* \sim B_{oa}$ relationship is crucial
for refining the energy partitioning process in the MEP model. The advancement of buoy observation
networks has provided compelling evidence for validating ocean heat fluxes and become crucial in assessing
their associated uncertainties (Bourras, 2006; Smith et al., 2011; Bentamy et al., 2017; Liang et al., 2022).
This study utilizes the energy balance-based MEP method to estimate ocean evaporation, introducing a novel
approach to redistributing surface energy budgets and offering a streamlined parameterization scheme
distinct from conventional bulk methods used for estimating ocean heat fluxes. In contrast to existing
approaches that using reanalysis-based schemes (e.g., NCEP, ECMWF, and GEOS) and their associated
parameterizations to estimate LE, this study employs satellite observations to directly estimate ocean heat
fluxes, thereby minimizing error propagation associated with the model structures and assimilation schemes.
Current global ocean surface heat flux datasets can be classified into five categories based on their
deriving approaches (Tang et al., 2023): remote sensing-based (e.g., J-OFURO3), atmospheric reanalysis-
based (e.g., ERA5), machine learning-based (e.g., OHFv2), in-situ based (e.g., NOC), and hybrid-based (e.g.,
OAFlux) approaches. Compared to terrestrial flux products, these ocean flux products generally have a
coarser spatial resolution ranging from 0.25º to 1.875º. Recent studies have conducted comprehensive
assessments of global ocean heat flux datasets regarding their accuracy and error characteristics across spatial
and temporal scales (Bentamy et al., 2017; Tang et al., 2023). However, substantial discrepancies remain
among these datasets, particularly in terms of spatial patterns, annual means, and interannual variabilities.
Therefore, developing a new global dataset using the innovative method could advance our understanding of
deriving algorithms, improve temporal and spatial coverage of flux variables with a higher accuracy, and
provide alternative reference to assess ocean surface heat fluxes in various applications. The primary
objectives of this study are as follows: (1) to develop and validate the MEP approach for estimating ocean
heat fluxes using observations from 129 stations; (2) to investigate the impact of heat storage on ocean energy
allocation and the influence of the Bowen ratio on energy partitioning for heat flux estimations; (3) to produce
a MEP-derived ocean heat fluxes product (spatial resolution: 0.25°; temporal coverage: 1988-2017) and
present its spatiotemporal patterns.
**2. Methods**
**2.1 Components of ocean surface energy balance**
The global ocean energy balance equation is as follows (Meehl, 1984; Wang et al., 2021):
$$R_n = LE + H + G \tag{1}$$

$$R_n = R_{ns} + R_{nl} = R_s^{\downarrow} - R_s^{\uparrow} + R_l^{\downarrow} - R_l^{\uparrow} \tag{2}$$

where $R_n$, $R_{ns}$, and $R_{nl}$ are net radiation, net shortwave radiation (the difference of incoming radiation
$R_s^{\downarrow}$ and reflected solar radiation $R_s^{\uparrow}$), and net longwave radiation (the difference of incoming longwave
radiation $R_l^{\downarrow}$ and outgoing longwave radiation $R_l^{\uparrow}$), respectively; $H$ is sensible heat, $LE$ is latent heat, and
$G$ is the heat flow through the surface. Unlike terrestrial surfaces, the energy balance equation for the ocean
surface accounts for distinct energy exchange processes, including the impact of seawater mixing and
dynamics on energy transfer. For the ocean surface, the flux term $G$ has two components,
$$G = G_t + G_v \tag{3}$$

where $G_t$ is the change in the ocean heat content ($\Delta OHC$, or heat storage), and $G_v$ is the lateral heat
transported by ocean currents and other processes. The $G_t$ can be quantified as the vertical integration of
temperature profile in a column of depth (Meehl, 1984, Li et al., 2023). Both the heat storage and the ocean
heat transport $G_v$ are difficult to quantify, which requires large masses of hydrographic variables and
performing integrations at different depths. Since the lateral heat transport by ocean currents is zero at the
global scale (Wang et al., 2021), $G$ can be regarded as equivalent to the change in ocean heat content or heat
storage at the global level. For the consistency throughout the paper, this study will consider the concept of
$G$ flux as equivalent to the changes in the heat storage.

## 2.2 The Maximum Entropy Production theory


2.2.1 The original MEP model
The MEP model simulates ocean surface heat fluxes using inputs variables of net radiation ($R_n$), surface skin
temperature ($T_s$), and surface specific humidity ($q_s$) under the constraint of the surface energy balance. The
latent heat, sensible heat, and surface thermal energy flux ($Q$) are calculated as,

$$[1 + B(\sigma) + \frac{B(\sigma)}{\sigma} \frac{I_s}{I_0} | H |^{-\frac{1}{6}}]H = R_n \tag{4}$$


$$E = B(\sigma)H \tag{5}$$


$$Q = R_{nl} - E - H \tag{6}$$


$$B(\sigma) = 6(\sqrt{1 + \frac{11}{36}\sigma} - 1), \quad \sigma = \frac{\lambda^2}{c_p R_v} \frac{q_s}{T_s^2} \tag{7}$$


$$I_0 = \rho_a c_p \sqrt{C_1 k z} (C_2 \frac{k z g}{\rho_a c_p T_r})^{\frac{1}{6}} \tag{8}$$


where $B(\sigma)$ is the reciprocal Bowen ratio, $\sigma$ is a dimensionless parameter that characterizes the phase
change at the ocean surface, $\lambda$ (J·kg$^{-1}$) is the latent heat of vaporization of liquid surface, $c_p$ ($10^3$ J·kg$^{-1}$·K$^{-1}$)
is the specific heat of air under constant pressure, and $R_v$ (461 J·kg$^{-1}$·K$^{-1}$) is the gas constant of water vapor.
$I_0$ is the "apparent thermal inertia" of air and describes the turbulent transport process of the boundary layer
based on the Monin-Obukhov similarity theory (MOST) (Wang and Bras, 2010). $I_s$ is the thermal inertia of
the ocean surface (J·m$^{-2}$·K$^{-1}$·s$^{-1/2}$), and can be parametrized as $I_s = \sqrt{\rho c \lambda}$ (with density $\rho$, the specific
heat $c$) represents the physical property of surface ($I_s = 1.56 \times 10^3$ J·m$^{-2}$·K$^{-1}$·s$^{-1/2}$ for water surface and
$1.92 \times 10^3$ J·m$^{-2}$·K$^{-1}$·s$^{-1/2}$ for ice surface).

Over the sea ice surface, assumed to be saturated, the specific humidity $q_s$ can be derived as a function

of surface temperature $T_s$ using the Clausius-Clapeyron equation. (El Sharif et al., 2019; Shaman & Kohn,

2009).

$$q_s = \varepsilon \frac{e_s(T_s)}{P} = \varepsilon \frac{e_0}{P} \exp[\frac{\lambda_s}{R_v}(\frac{1}{T_0} - \frac{1}{T_s})] \tag{9}$$

where $\varepsilon$ (= 0.622) represents the ratio of the molecular weight of water vapor to that of dry air, $e_s(T_s)$
denotes the saturation vapor pressure at temperature Ts, $e_0$ is the saturation vapor pressure at the reference
temperature $T_0$ (273.15 K), and $P$ is the atmospheric pressure (mb).
### 2.2.2 Specific improvements on the MEP model
According to the MEP theory, the net solar radiation ($R_{ns}$) entering the water surface medium is absorbed by
the water body, with the allocable radiation flux denoted as $R_{nl} = E + H + Q$ (Eq.6). Consequently, the
expression for ocean heat uptake (heat storage) is derived as $G = R_n - E - H = R_{ns} + Q$. While this theory
has received preliminary validation in shallower water bodies, such as lake surfaces (Wang et al., 2014), its
applicability on deeper water bodies with larger heat storage capacities in ocean surfaces requires further
evaluation. This study introduced two key hypotheses: (1) The substantial heat storage capacity of the ocean
could exert a significant influence on seasonal latent and sensible fluxes, potentially introducing bias to the
MEP equations, (2) The notable underestimation of latent heat flux and overestimation of sensible heat flux
by the MEP model point to a significant deviation from the Bowen's ratio formula, necessitating a reasonable
correction. To address this, the study proposed two approaches for enhancing the MEP formulas: (1)
Considering the impact of heat storage in the MEP's energy balance equation, and (2) Adjusting the
theoretical equilibrium Bowen ratio within the MEP model. This can be specifically represented as follows:

$$[1+\frac{1}{B_o^*}]H = R_n - G \tag{10}$$

$$B_o^* = \frac{1}{B(\sigma)} \tag{11}$$

$$B_{oa} = a \times B_o^* + b \tag{12}$$

where $B_o^*$ is the equilibrium Bowen ratio, which denotes the theoretical ratio of sensible heat flux to latent heat flux when the surface and the atmosphere are in equilibrium regarding water vapor. Correspondingly, the corresponding evaporation at this condition is known as equilibrium evaporation (defined as the water vapor evaporating from a saturated surface into a saturated atmosphere). To accurately predict actual evaporation, a reliable functional relationship needs to be established to predict $B_{oa}$ from $B_o^*$. Empirical studies have introduced coefficients to correlate $B_o^*$ to $B_{oa}$ under diverse environmental circumstances; for instance, the Priestley–Taylor coefficient was expressed as (Priestley & Taylor, 1972).

$$B_{oa} = 0.79 \times B_o^* - 0.21 = \frac{0.79 - 0.21 \times B(\sigma)}{B(\sigma)} \tag{13}$$

Further studies have led to the emergence of more updated empirical coefficients. Hicks and Hess (1977) estimated the actual Bowen ratio as $B_{oa} = 0.63 \times B_o^* - 0.15$ by aligning it with direct observations of the fluxes. Yang & Roderick (2019) deduced an empirical coefficient of 0.24 and formulated it as $B_{oa} = 0.24 \times B_o^*$ through fitting Bowen ratio and surface temperature data across the global ocean surface. Furthermore, Liu & Yang (2021) derived a new equation as $B_{oa} = 0.37 \times B_o^* - 0.05$ based on the atmospheric boundary layer model. Given their favorable spatial applicability and representativeness, this study opted to utilize these four $B_{oa} \sim B_o^*$ formulas to refine the MEP model and assess their suitability. The revised reciprocal actual Bowen ratio was represented as,

$$\left\{ \begin{array}{l} B(\sigma)_{a1} = \dfrac{1}{B_{oa}} = \dfrac{B(\sigma)}{0.79 - 0.21 \times B(\sigma)} \\[3mm] B(\sigma)_{a2} = \dfrac{1}{B_{oa}} = \dfrac{B(\sigma)}{0.63 - 0.15 \times B(\sigma)} \\[3mm] B(\sigma)_{a3} = \dfrac{1}{B_{oa}} = \dfrac{B(\sigma)}{0.24} \\[3mm] B(\sigma)_{a4} = \dfrac{1}{B_{oa}} = \dfrac{B(\sigma)}{0.37 - 0.05 \times B(\sigma)} \end{array} \right\} \tag{14}$$

where $B(\sigma)_{a1} \sim B(\sigma)_{a4}$ represent the four empirical Bowen ratio formulas for comparisons in this study.
Thus, the workflow of the improved MEP model was conducted as: replacing the original $B(\sigma)$ with
the corrected $B(\sigma)_a$, then combining Eq. (5), (7)-(9), and (14) into the MEP energy balance equation
considering heat storage (Eq. (10)), ultimately leading to the determination of latent and sensible heat flux.

### 2.3 Sensitivity analysis

To quantify the influence of input variables in the MEP model on evaporation estimate at the ocean surface,
the sensitivity coefficient (S) was computed as (Beven, 1979; Isabelle et al., 2021),

$$S_i = \frac{\partial LE}{\partial x_i} \frac{x_i}{LE} \tag{15}$$

where $S_i$ represents the sensitivity coefficient of $LE$ to each variable $x_i$. The magnitude of $S_i$ reflects the degree
of impact of the variable's changes on LE; a larger absolute value indicates a greater influence of the variable
on $LE$. A positive value signifies a positive correlation between evaporation and the variable's changes, while
a negative value indicates a negative correlation. For example, a sensitivity coefficient of 0.5 represents that
a 10% increase in the variable would result in a 5% increase in $LE$. The sensitivity levels can be categorized
based on the absolute value |Si| (Lenhart et al., 2002; Yin et al., 2010): |Si| > 1 indicates very high sensitivity,
1 > |Si| > 0.2 denotes high sensitivity, 0.2 > |Si| > 0.05 reflects moderate sensitivity, and |Si| < 0.05 suggests
negligible sensitivity.

### 2.4 Data fusion methods

To drive the improved MEP model with high-quality input data, this study aims to obtain a heat storage
dataset with optimal accuracy. The accuracy of the heat storage dataset was assessed using three approaches:
(1) individual dataset, (2) a fused dataset generated using the Bayesian Three-Cornered Hat (BTCH) method
(He et al., 2020), and (3) an ensemble means obtained through the arithmetic average (AA) method. Previous
studies have demonstrated that the BTCH method effectively quantifies uncertainties across diverse datasets
and improves accuracy by integrating multiple datasets without requiring prior knowledge (Long et al., 2017;
Liu et al., 2021; Duan et al., 2024). A recent study further evaluated various data fusion methods, including
BTCH and the AA method, for addressing uncertainties in global evapotranspiration estimates derived from
different datasets. The findings revealed that while both BTCH and AA are effective in identifying lower-
quality ET datasets, their ability to consistently produce higher-accuracy datasets remains uncertain and, in
some cases, may even degrade the overall accuracy (Shao et al., 2022). The performance of these fusion
methods is highly sensitive to the selection of input datasets. For instance, the AA method is particularly
susceptible to the influence of lower-quality datasets, especially when the sample size is small. Similarly, the
performance of BTCH diminishes as the error covariance among the included datasets increases.
Consequently, following a comparative analysis of the accuracy of individual, BTCH, and AA fusion datasets,
this study selected the optimal heat storage dataset to drive the MEP model. Since BTCH is not the primary
focus of this study, detailed method descriptions are referred to He et al. (2020).
**3. Data materials**
**3.1 Input data for MEP model**
The performance of both the original and improved Maximum Entropy Production (MEP) models was
evaluated using observed data from in-situ buoy stations, as described in Section 3.2. The optimal empirical
Bowen ratio formula for the MEP model was then determined through multi-site assessments. Subsequently,
the improved MEP model was applied to estimate global heat fluxes using long-term remote sensing data, as
detailed in Sections 3.3 and 3.4. Specifically, the input variables of net radiation, heat storage, and sea surface
temperature driving the improved MEP model were derived from the J-OFURO3 dataset, spanning 1988 to
2017 with a spatial resolution of 0.25°, as outlined in Section 4.3.
**3.2 In situ buoy observations**
A total of 129 in situ buoy sites were employed for ocean heat fluxes calculation and validation with MEP
model and its modified version, as listed in Table 1. About 96% of selected sites (124 of 129 all sites) were

collected from the Global Tropical Moored Buoy Array (available at https://www.pmel.noaa.gov/), which

consists of the TAO/TRION Pacific Ocean (69 buoys), PIRATA Atlantic Ocean (23 buoys), and RAMA

Indian ocean (32 buoys), and the remaining sites including Upper Ocean Processes Group (3 buoys) consists

of Project WHOTS - WHOI Hawaii Ocean Time-series Station (available at

https://uop.whoi.edu/ReferenceDataSets/whotsreference.html), Project NTAS - Northwest Tropical Atlantic

Station and Project STRATUS (https://uop.whoi.edu/ReferenceDataSets/ntasreference.html), and the Pacific

ocean climate stations (2 buoys) consists of KEO and PAPA moorings

(https://www.pmel.noaa.gov/ocs/data/fluxdisdel/). The availability of all buoy stations refers to the "Data

availability" section. The observational sites covered the spatial range of 25°S~ 50.1°N latitude, temporal

range span from 1989/12 to 2023/12. Observational meteorological variables and heat fluxes included the

net longwave radiation, net shortwave radiation, sea surface skin temperature, specific humidity at 2m height

(if available, or computed as the function of SST according to the Clausius‐Clapeyron equation), latent heat

flux and sensible heat flux. Limited by the availability of longwave radiation observations, the net radiation

had a relatively shorter time series length compared to latent and sensible heat fluxes. The surface air-sea

fluxes of buoy observations were computed using the COARE 3.0b algorithm, which have been widely

applied for fluxes estimations and validations (Tang et al., 2023; Bentamy et al., 2017; Fairall et al., 2003).

All the selected original buoy observation (except for KEO and Papa sites) records were in monthly temporal

resolution, and the original daily observations of KEO and Papa had been aggregated to monthly by simple

average method. The spatially distributed map of all selected sites was illustrated in Fig.S1.

**Table 1.** Information about observational ocean surface heat fluxes of 129 buoy sites

| Buoy array | Buoy amount | Spatial coverage | Temporal coverage | Number of $LE$ ($H$) records | Number of $R_n$ records |
|---|---|---|---|---|---|
| TAO/TRION pacific | 69 | 165°E-95°W 10°S-10°N | 1989/12/16- 2023/12/16 | 12377 | 522 |
| PIRATA Atlantic | 23 | 40°W-10°E, 20°S-20°N | 1997/9/16- 2023/12/16 | 2644 | 631 |
| RAMA Indian | 32 | 55°E-100°E, 25°S-15°N | 2001/11/16- 2023/12/16 | 1862 | 286 |
| WHOTS | 1 | 158°W, 22.7°N | 2004/08/15- 2021/08/15 | 205 | 205 |

| NTAS | 1 | 51°W, 15°N | 2001-04/15-2020/03/15 | 219 | 219 |
| STRATUS | 1 | 85.4°W, 19.6°S | 2000/10/15-2021/01/15 | 235 | 235 |
| KEO | 1 | 144.6°E, 32.3°N | 2004/06/17-2023/08/12 | 177 | 177 |
| PAPA | 1 | 144.9°W, 50.1°N | 2007/06/08-2023/11/14 | 181 | 181 |

Note: The number of records represents the effective count (excluding NA values) of latent and sensible heat flux observations.

### 3.3 Global turbulent heat flux datasets for evaluations

This study evaluated and compared 7 global turbulent heat flux products with observations, categorizing them into three types: Remote sensing-based, atmosphere reanalysis-based, and hybrid-based (Table 2). These seven products encompassed monthly data spanning from 1988 to 2017, with spatial resolutions ranging from 0.25° to 1°. The criterion for dataset filtering prioritized products with relatively long time series, typically exceeding 15 years.

The Clouds and Earth's Radiant Energy Systems Synoptic Edition 4A (CERES SYN1deg_Ed4A, hereafter referred to as CERES4, available at https://ceres.larc.nasa.gov/data/) offers net radiation data, derived from clear-sky upward shortwave, downward shortwave flux, upward longwave, and downward longwave flux measurements (Wielicki et al., 1996; Rutan et al., 2015). Another remote sensing-based radiation product, the Global Energy and Water Cycle Experiment - Surface Radiation Budget (GEWEX-SRB, available at https://asdc.larc.nasa.gov/project/SRB) (Pinker et al., 1992), in conjunction with CERES4, demonstrated good accuracy in retrieving $R_n$, as validated by six global observing networks (Liang et al., 2022).

The J-OFURO3 is the third-generation dataset developed by the Japanese Ocean Flux Data Sets with use of the Remote-Sensing Observations (J-OFURO) research project (available at https://j-ofuro.isee.nagoya-u.ac.jp/en/) (Tomita et al., 2019). It calculated turbulent heat flux with the latest version of COARE3.0 algorithm, and provided datasets for $R_n$, *LE, H* and *SST* in this study. Validation with in situ observations showed that J-OFURO3 offered a superior performance of latent heat compared to other 5 satellite products from 2002-2013 (Tomita et al., 2019).

Two Atmosphere reanalysis products including the fifth generation European Centre for Medium-Range
Weather Forecasts (ECMWF) atmospheric Re-Analysis52 (ERA5, available at
https://cds.climate.copernicus.eu/cdsapp#!/dataset/reanalysis-era5-single-levels-monthly-
means?tab=overview) (Hersbach et al., 2020), and the Modern-Era Retrospective analysis for Research and
Applications Version2 (MERRA2, available at
https://disc.gsfc.nasa.gov/datasets/M2TMNXOCN_5.12.4/summary) (Gelaro et al., 2017). Both ERA5 and
MERRA2 products employed the bulk formula based on the MOST to calculate heat fluxes. Validation
results from previous studies have demonstrated good consistency with buoy estimates regarding heat fluxes
(Pokhrel et al., 2020; Chen et al., 2020).
The OAFlux (available at https://oaflux.whoi.edu/), a hybrid-based product developed under the
Objectively Analyzed Air-Sea Fluxes (OAFlux) project at the Woods Hole Oceanographic Institution (WHOI)
(Yu et al., 2008), was utilized for comparisons with ocean heat fluxes derived from distinct methods. This
product calculates fluxes based on the COARE3.0 bulk algorithm and employs a variational objective
analysis to determine the optimal fitting of independent variables. Detailed descriptions on all utilized global
turbulent heat fluxes products, and their validation performances against buoy observations with reported
studies were available in Tang et al (2023).

**Table 2.** The information of the 7 used global radiation and heat fluxes products

| Product | Variables | Spatial resolution | Time span | Type | Reference |
|---|---|---|---|---|---|
| CERES4 | $R_n$ | 1° | 2000-2017 | Remote sensing | Rutan et al. (2015) |
| GEWEX-SRB | $R_n$ | 1° | 1988-2017 | Remote sensing | Pinker et al. (1992) |
| J-OFURO3 | $R_n$, $SST$, $LE$, $H$ | 0.25° | 1988-2017 | Remote sensing | Tomita et al. (2019) |
| ERA5 | $R_n$, $LE$, $H$, $P$ | 0.25° | 1988-2017 | Atmosphere reanalysis | Hersbach. et al. (2020) |
| MERRA2 | $R_n$, $LE$, $H$ | $1/2° \times 2/3°$ | 1988-2017 | Atmosphere reanalysis | Gelaro et al. (2017) |
| OAFlux | $LE$, $H$ | 1° | 1988-2017 | Hybrid-based | Yu et al. (2008) |
| IAPv4-OHC | $OHC$ | 1° | 1988-2017 | Hybrid-based | Cheng et al. (2017) |

**3.4 Ocean heat content data**

Remote sensing data for heat storage ($G$) was primarily derived from two categories: the first category included data obtained from the residual of the energy balance equation ($R_n$-$LE$-$H$), including J-OFURO3, ERA5, and MERRA2; the second category was calculated from changes in Ocean Heat Content (OHC). The ocean heat content data was obtained from IAP OHC gridded analysis (IAPv4, available at http://www.ocean.iap.ac.cn/) dataset, covering ocean depth of 0-6000m (Cheng et al., 2017), and has been extensively utilized in global ocean heat analysis, ocean warming, and climate change studies (Li et al., 2023; Cheng et al., 2022; Cheng et al., 2024). The delta OHC was calculated using the numerical differentiation method (Xu et al., 2019) as $\Delta\text{OHC}(i) = \dfrac{OHC(i+1)-OHC(i-1)}{2\Delta i}$, $i$ denotes the OHC of $i$-th month.

At the WHOTS site, this study compared the OHC changes at different depths with the observed $G$, derived as $Rn$-$LE$-$H$ (Table S1). Since the OHC variation from 0~100m depth exhibited the smallest error with the observations, the data from 0~100m depth range were chosen as the heat storage. This study assessed the suitability of $G$ flux and $\Delta\text{OHC}$ for global evaporation estimations, with the aim of minimizing the errors introduced by input variable data in the MEP model.

This study evaluated the accuracy of all the variables $R_n$, $T_s$, and $G$ using the aforementioned datasets on a global scale by comparing them against buoy observations (in Section 4.3), to optimize input accuracy for driving the MEP model. To maintain consistency in the analysis, this study resampled all products to 1° spatial resolution when comparing the Bowen ratio across multiple products. Nevertheless, when conducting site validations with buoy observations, the original resolution of the data was preserved to minimize uncertainty attributable to scale effects.

**4. Results**

**4.1 The new MEP model with heat storage and the revised Bowen ratio formulas**

To demonstrate how the MEP model has been developed and improved, we showed the comparisons of different MEP models in simulating heat fluxes across 129 global buoy stations (Fig.1). Limited by the availability of $R_{nL}$ data, we used $LE + H$ instead of the available energy ($R_n - G$), enabling the utilization of more observational records to verify the MEP model. The original MEP model (without considering heat

storage) showed a significant negative correlation between $LE$ and $H$ (with $R^2$ exceeding 0.65 as in Fig.1a &
Fig.1c), with considerable errors, where the RMSE of $LE$ was 134.6 W·m$^{-2}$ and that of $H$ was 37 W·m$^{-2}$.
After incorporating the influence of heat storage effects (represented as $MEP\_M$, as depicted in Fig.1b and
Fig.1h), the MEP-simulated $LE$ showed a good consistency with buoy observations, with an $R^2$ value of 0.97
and a reduced RMSE of 27 W·m$^{-2}$. However, the $MEP\_M$ method revealed a significant bias in the
partitioning of $LE$ and $H$ from the available energy. Specifically, $LE$ was underestimated by 25% (regression
slope = 0.75), while H was overestimated by 46% compared to observations. This finding agreed with
previous research findings that equilibrium evaporation tended to underestimate actual evaporation from
saturated surfaces by 20%~30% (Yang & Roderick, 2019; Philip, 1987). The significant difference between
$B_{oa}$ and $B_{o*}$ could exist as the equilibrium evaporation is considered as the lower limit of actual evaporation
from saturated surfaces (Priestley and Taylor, 1972). To address the deviation between $B_{oa}$ and $B_o^*$, it is
necessary to convert the equilibrium Bowen ratio into the actual Bowen ratio, allowing for a more reasonable
and accurate allocation of surface energy budget.

After incorporating the effects of heat storage, four variants of the MEP model were developed by

replacing $B_{o*}$ with $B_{oa}$ derived from four different empirical formulas. These variants were defined as follows:
$M\_0.24$ (where $B_{oa}=0.24 B_o^*$), $M\_0.79$ (where $B_{oa}=0.79 B_o^*-0.21$), $M\_0.63$ (where $B_{oa}=0.63B_o^*-0.15$), and
$M\_0.37$ (where $B_{oa}=0.37B_o^*-0.05$). Adjusting the Bowen ratio significantly improved the accuracy of the
energy flux estimates. The simulated $LE$ exhibited strong agreement with observations, with all $R^2$ exceeding
0.97 and RMSE ranging from 4.7 W·m$^{-2}$ (for $M\_0.24$) to 7.1 W·m$^{-2}$ (for $M\_0.79$), which was lower than that
derived from $B_o^*$ (RMSE = 27 W·m$^{-2}$). Both $M\_0.79$ and $M\_0.63$ tended to underestimate $LE$, especially
when $LE$ exceeded 200 W·m$^{-2}$ (Fig. 1d and Fig.1e). For the simulated $H$, the $M\_0.24$ outperformed the other
three, showing the smallest errors and highest $R^2$.


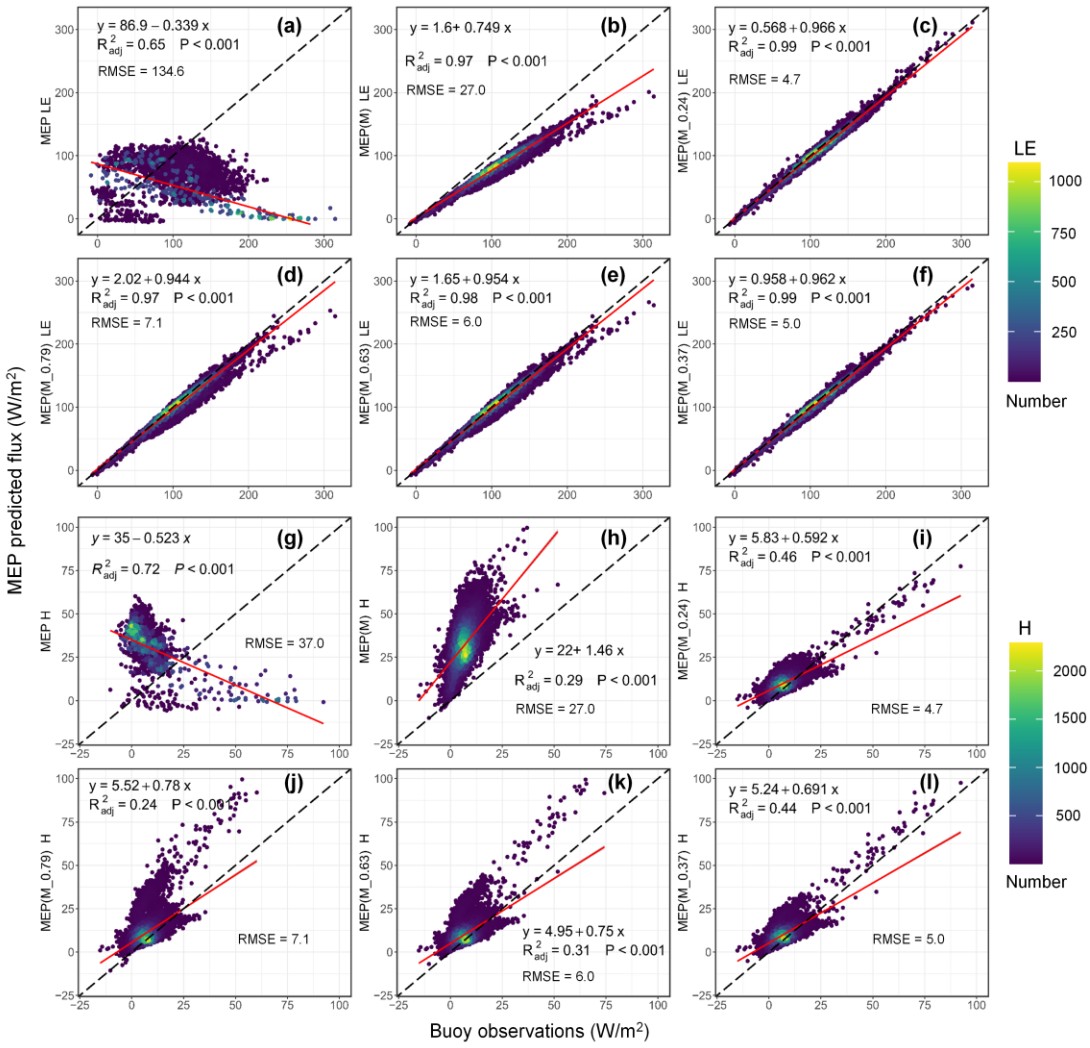

**Figure 1.** Scatter density plots of monthly latent heat flux (a~f) and sensible heat flux (g~l) derived by the
original and modified MEP methods versus observations from 129 buoy stations (as in Table 1). (a) The
original MEP method, (b) The modified MEP method considering the heat storage effect, (c) The modified
MEP method considering both the heat storage and empirical Bowen ratio formula $B_{oa}=0.24B_o^*$, (d)~(f) for
the modified MEP method considering both the heat storage and empirical Bowen ratio formulas
$B_{oa}=0.79B_o^*-0.21$, $B_{oa}=0.63B_o^*-0.15$, and $B_{oa}=0.37B_o^*-0.05$. (g)~(l) are the same with (a)~(f) but for sensible
heat flux.

Specifically, the spatial patterns of simulated errors for the four variants of the MEP model were

obtained (Fig. 2), along with the errors across different observational buoy arrays (Fig. 3). Overall, the four

variants of the improved MEP models demonstrated relatively lower bias at low latitudes (10°S to 10°N) but

exhibit larger bias in higher latitude regions (above 15°N), particularly at the KEO, WHOTS, and STRATUS

buoy sites. Comparing the four formulas across varying latitudes, the *M_0.24* formula exhibited the smallest

RMSE (ranging from 3.6 to 12 W·m$^{-2}$) (Fig. 3c), while the *M_0.79* formula showed the largest errors (RMSE

ranging from 3.9 to 26.6 W·m$^{-2}$). This consistency was also evident in the Kling-Gupta Efficiency (KGE)
coefficient, with *M_0.24* demonstrating superior performance in terms of accuracy, robustness, and
adaptability. In term of *M_0.24* formula, the prediction errors across observational arrays ranked as follows:
RAMA < PIRATA < TAO/TRION < PaPa < KEO < STRATUS < WHOTS < NTAS. The arrays with
relatively larger RMSE (NTAS in the Atlantic Ocean, WHOTS, and STRATUS in the Pacific Ocean) may
originate from the larger observed values of LE (Fig. S2).

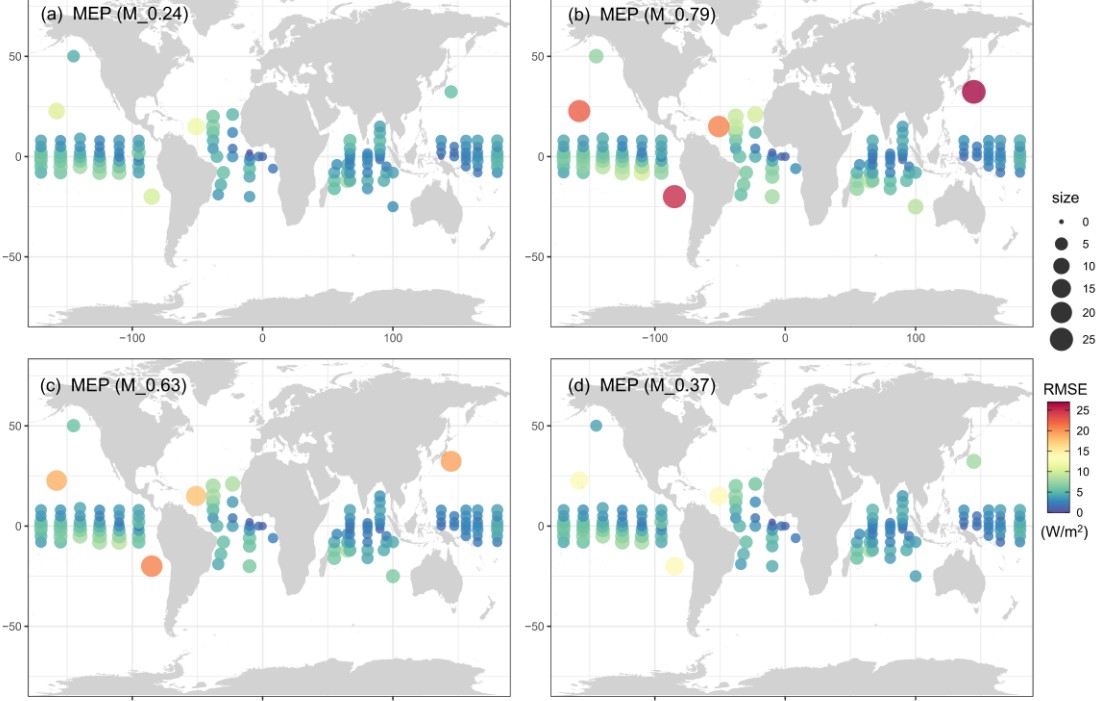

**Figure 2.** Spatial distribution of RMSE values in the comparison between latent heat flux estimated by the
improved MEP method (modified by four different Bowen ratio formulas) and buoy observations from 129
stations.

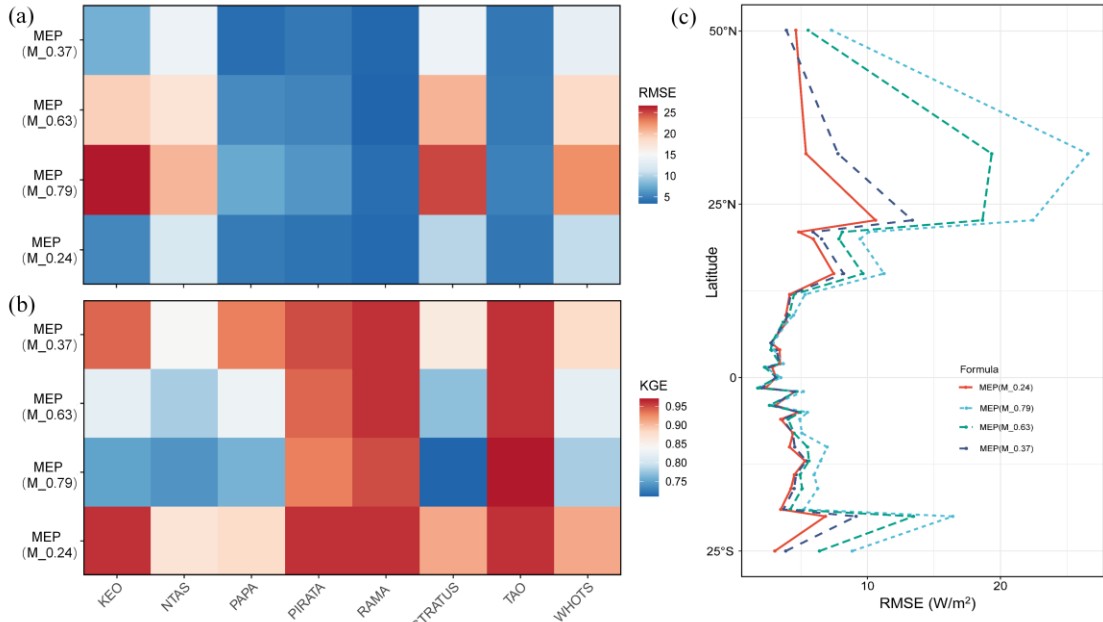


**Figure 3.** Comparisons between latent heat flux estimated by the improved MEP method using four empirical
Bowen ratio formulas and the buoy observations from each buoy array in term of RMSE (a) KGE value (b),
and latitudinal means of RMSE (c). Latitudinal means are based on data from 129 available buoy sites.

## 4.2 Dynamics of heat fluxes and Bowen ratio between original and improved MEP model

To thoroughly investigate the role of heat storage in the partitioning of surface energy and its implications
for the temporal dynamics of heat fluxes, we selected the KEO site for detailed analysis. This selection was
based on the site's long-term observational records and notable variability in flux patterns, which offered an
ideal context for a rigorous comparison of model-simulated error margins.
The improved MEP methods demonstrated comparable performance in estimating heat fluxes at the
KEO site when compared with other 128 sites (Fig. S3, Fig. 1), with the MEP (M_0.24) model exhibited the
most effective performance. Analysis of the time series data revealed significant variations in latent heat,
sensible heat, and Bowen ratio (Fig. 4). In the original MEP theory, the estimated *LE* exhibited an opposite
variation cycle (peak versus trough) compared to the observations. For instance, over a yearly period, the
observed peak in *LE* occurred in January 2005 (269 $W \cdot m^{-2}$) and the trough in June 2005 (6.9 $W \cdot m^{-2}$). In
contrast, the MEP simulated the peak in *LE* to occur in August 2005 (105 $W \cdot m^{-2}$) and the trough in December
2004 (0.7 $W \cdot m^{-2}$), resulting in a phase difference of 7 months for the peak and 6 months for the trough values.
Sensible heat flux (Fig. 4b) showed similar phase differences: observed *H* peaked in January 2005 (79 $W \cdot m^{-}$
$^2$) and reached its minimum in June 2005 (-3 W·m$^{-2}$), whereas MEP simulated $H$ to peak in August 2005 (46
W·m$^{-2}$) and reach its minimum in December 2004 (0.6 W·m$^{-2}$), consistent with the pattern observed for $LE$.
It was noteworthy that the original MEP model simulated variations in $LE$ and $H$ align with $R_n$ (Fig. S4),
which was reasonable over land where the small $G$ value can often be disregarded. However, over the ocean,
the observed variations in $R_n$ and $LE$ do not align in terms of their cycles. The maximum $R_n$ occurred in June
2004 (329 W·m$^{-2}$) and the minimum occurred in December 2004 (142 W·m$^{-2}$), with a 6-month delay in
relation to the variations in $LE$. Specifically, the peak $R_n$ corresponded to the trough of $LE$, and the trough $R_n$
corresponded to the peak of $LE$. This delay indicated that the heat storage effect delayed the peak of $LE$ and
altered the seasonal variations of $LE$ and $H$.

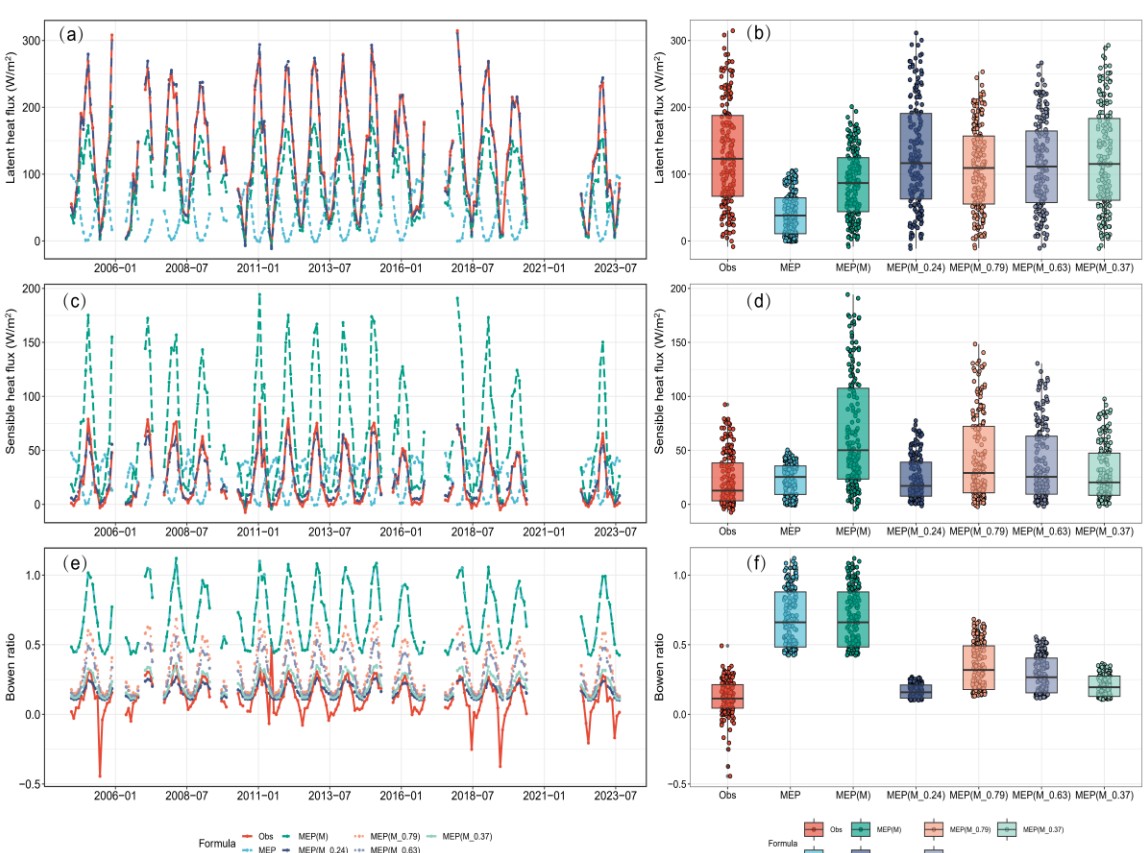


**Figure 4.** The inter-annual variations (a, c and e) and variabilities (b, d, and f) for latent heat flux, sensible
heat flux, and Bowen ratio at KEO site from June 17, 2004, to August 12, 2023. The fluxes in the comparison
include observations and the estimates from MEP using the original formula (MEP), the formula
incorporating the ocean heat storage (MEP(M)), and four other formula considering both ocean heat storage
and adjustment of Bowen ratio. Note that the (a) and (c) only display results using MEP *(M_0.24)* among all
four empirical Bowen ratio formulas for clearer comparison.

For the patterns of the Bowen ratio, both the original MEP formula and the modified formulas exhibited

consistent patterns with the observed values. The observed maximum Bowen ratio occurred in January 2005

(0.29) and the minimum in June 2005 (-0.4). However, the original MEP formula simulated a maximum of

1.01 and a minimum of 0.44, indicating a significant overestimation compared to the observed Bowen ratio.

This discrepancy suggested that on the ocean surface, the available energy ($R_n$-$G$) was predominantly

allocated to *LE* (Fig.S4). Among the four empirical formulas, *M_0.24* simulated *LE*, *H*, and Bowen ratio

values were closest to the observed values. The median of the observed Bowen ratio was 0.11, while the

original MEP Bowen ratio was 0.66. Among the four modified Bowen ratio formulas (*M_0.24*, *M_0.79*,

*M_0.63*, *M_0.37*), their median Bowen ratios were 0.15, 0.32, 0.27, and 0.19 respectively, with *M_0.24* being

the closest to the observed Bowen ratio.

Heat storage is crucial for the energy distribution process over the ocean surface. While the original

MEP formulas have been effectively validated when applied to surfaces with shallow depths such as water

and snow (Wang et al., 2014), they exhibit significant uncertainty when applied to the ocean surface. This

discrepancy primarily arises from the fact that land is a non-transparent medium with relatively small heat

storage values at monthly scales. Similarly, shallow water bodies also exhibit small heat storage values that

can often be ignored. In the study by Wang et al. (2014), for example, two lakes with depths of 2m (Lake

Tämnaren) and 4m (Lake Råksjö) still resulted in underestimated *LE*. However, for deeper lakes (generally >

3m depth), heat storage becomes significant and cannot be neglected (Zhao et al., 2016; Zhao & Gao, 2019).

On deep ocean surfaces, with the most recent average depth estimate of 3,682m from NOAA satellite

measurements, heat storage variations can influence depths up to 6,000m (Cheng et al., 2017). Therefore, the

impact of heat storage was substantial and cannot be disregarded. In the original MEP theory, heat storage

was not considered in the energy balance equation, where it was assumed that the net solar radiation ($R_{ns}$) is

absorbed by the ocean and $R_{nL}$= $LE + H + Q$. Then, the heat storage was obtained as $G = R_{ns} + Q$. In this

study, we compared the characteristics of MEP-derived G ($R_{ns} + Q$) with the observed *G* flux ($G = R_n - LE$

$- H$ (Fig. S5). MEP-derived *G* showed a good correlation ($R = 0.96$) and consistent trends with the observed

values (Fig. S5a & b), ranging from -4 to 81 W·m$^{-2}$. However, MEP-calculated *Q* (ranged from -210 to -65

W·m$^{-2}$) exhibited a negative correlation with the observed *G* (which ranged from -386 to 200 W·m$^{-2}$). Both

MEP-derived *G* and *Q* fluxes were significantly underestimated. Therefore, the prediction errors in *LE* and

*H* originated from the inability to accurately quantify the heat storage. Hence, considering the influence of
heat storage was crucial for accurately predicting *LE* and *H* over the ocean surface.
**4.3 Evaluation of global radiation and heat storage flux**
4.3.1 Evaluation of net radiation
After considering the effect of heat storage and the Bowen ratio, the improved MEP method demonstrated
its high performance at the site scale. The results suggested that improved MEP method held substantial
promise for further application at a global scale. To facilitate this, we assessed the primary input variables of
the improved MEP method (including $R_n$, $G$, and $T_s$) to identify datasets with the best accuracy.
Net radiation, as the primary variable in the energy balance equation, significantly influenced the
uncertainty of the MEP model (Huang et al., 2017). Selecting a reliable $R_n$ product was essential for accurately
estimating global latent and sensible heat fluxes. Previous studies have evaluated the available global ocean
surface $R_n$ datasets at daily scales using observations from 68 moored buoy sites (Liang et al., 2022). In this
study, we conducted a comprehensive evaluation of current available monthly $R_n$ products, including three
remote sensing-based products (CERES4, GEWEX-SRB, and JOFURO3) and two atmosphere reanalysis-
based products (ERA5 and MERRA2) at 129 buoy sites. All products exhibited good consistency with buoy
observations (Table 3 and Fig. S6), with $R^2$ values greater than 0.78. In terms of RMSE, the error rankings
for all products were as: J-OFURO3 (10 W·m$^{-2}$) < ERA5 (39.03 W·m$^{-2}$) < CERES4 (40.67 W·m$^{-2}$) <
GEWEX-SRB (41.83 W·m$^{-2}$) < MERRA2 (49.23 W·m$^{-2}$). It was evident that J-OFURO3 demonstrated the
highest accuracy, as indicated by RMSE, NSE, and KGE statistics. This result was also consistent with
previous assessments of global $R_n$ (Liang et al., 2022), emphasizing J-OFURO3 as the least erroneous among
all individual products and superior to existing alternatives including CERES4, ERA5, MERRA2, GEWEX-
SRB, JRA55, OAFlux, and TropFlux.
**Table 3.** Evaluation of global monthly net radiation products against buoy observations

| Products | $R^2$ | ME (W·m$^{-2}$) | MAE (W·m$^{-2}$) | RMSE (W·m$^{-2}$) | PBIAS (%) | NSE | KGE |
|---|---|---|---|---|---|---|---|
| J-OFURO3 | **0.96** | **1.6** | **7.3** | **10.0** | **1.0** | **0.96** | **0.97** |
| ERA5 | 0.79 | 28.8 | 30.3 | 39.0 | 17.8 | 0.45 | 0.77 |
| MERRA2 | 0.78 | 39.7 | 41.2 | 49.2 | 24.8 | 0.15 | 0.68 |
| CERES4 | 0.81 | 31.4 | 32.6 | 40.6 | 19.6 | 0.42 | 0.76 |
| GEWEX-SRB | 0.78 | 32.6 | 33.8 | 41.8 | 20.2 | 0.37 | 0.76 |

Note: The evaluation period for all datasets is 1988-2017, except for CERES4, which spans from March 2000 to December 2017. The best-performed statistics are indicated in bold type.

### 4.3.2 Evaluation of heat storage

The study underscored the importance of considering heat storage in simulating heat fluxes using the improved MEP model. For the first time, we assessed global heat storage using the J-OFURO3, ERA5, MERRA2, and ΔOHC datasets. In addition to assessing these individual datasets, we investigated the potential for enhancing accuracy through data fusion methods. We employed the BTCH and AA method to fuse heat storage data and compared the accuracy between individual datasets and fused datasets (Table 4). The results revealed that while using the AA method (e.g., AA4) to fuse yields smaller errors compared to ERA5, MERRA2, and ΔOHC, it still failed to achieve the accuracy of the J-OFURO3 product. Similarly, the BTCH method, despite fusing data from three or four sources, also does not match the accuracy of the J-OFURO3 method, as indicated by metrics of $R^2$, RMSE, and KGE. The heat storage derived from J-OFURO3 data showed high consistency with observations ($R^2$=0.95), as illustrated in Fig. 5 (spatial distribution of errors depicted in Fig.S7). Therefore, this study employed the heat storage data derived from the J-OFURO3 dataset as the input for the MEP model.

To ensure consistency with radiation data source, the Sea Surface Temperature ($SST$) data from J-OFURO3 was utilized for $T_s$ inputs, which was derived as the ensemble median from 12 global $SST$ products (Tomita et al., 2019). Ultimately, the input variables including net radiation, heat storage, and sea surface temperature for driving MEP model were all determined from the J-OFURO3 dataset spanning from 1988 to 2017. Saturated specific humidity was computed as a function of $SST$ and surface air pressure (from ERA5) using the Clausius-Clapeyron equation. The reliability of gridded data for the variables $R_n$, $G$, and $T_s$ were simultaneously examined at an observational site (Fig.S8), where all three variables demonstrated high consistency with observed data from August 2004 to December 2017 (with $R^2 > 0.96$), effectively capturing the monthly dynamics of $R_n$, $G$, and $T_s$.

**Table 4.** Assessment of monthly heat storage between global remote sensing datasets and buoy observations

| Products | $R^2$ | ME (W·m$^{-2}$) | MAE (W·m$^{-2}$) | RMSE (W·m$^{-2}$) | PBIAS (%) | NSE | KGE |
|---|---|---|---|---|---|---|---|
| J-OFURO3 | **0.95** | **-3.5** | **15.3** | **19.7** | **-7.4** | **0.94** | **0.91** |
| ERA5 | 0.88 | 7.0 | 25.1 | 33.2 | 14.8 | 0.84 | 0.80 |

| | | | | | | | |
|---|---|---|---|---|---|---|---|
| MERRA2 | 0.86 | 11.6 | 27.1 | 36.1 | 24.5 | 0.81 | 0.72 |
| OHC | 0.35 | -48.2 | 64.4 | 86.9 | -101.9 | -0.11 | -0.10 |
| BTCH3-1 (E M J) | 0.89 | 7.1 | 22.8 | 30.5 | 15.1 | 0.86 | 0.81 |
| BTCH3-2 (E M O) | 0.88 | 4.6 | 24.0 | 31.4 | 9.9 | 0.85 | 0.86 |
| BTCH4 | 0.91 | 5.9 | 19.7 | 26.2 | 12.5 | 0.90 | 0.86 |
| AA2(EM) | 0.87 | 9.3 | 25.4 | 34.1 | 19.7 | 0.83 | 0.76 |
| AA3 (EMJ) | 0.91 | 4.7 | 20.2 | 26.7 | 10.1 | 0.90 | 0.87 |
| AA4 (E M J O) | 0.91 | 11.5 | 21.4 | 28.6 | 24.4 | 0.88 | 0.74 |

Note: BTCH3-1 represents the fusion of three products (ERA5, MERRA2, and J-OFURO3) using the BTCH method; TCH3-2 represents the fusion of ERA5, MERRA2, and OHC; BTCH4 represents the fusion of ERA5, J-OFURO3, MERRA, and OHC. AA denotes the Simple Arithmetic Average (AA) method. The evaluation period spans from 1988 to 2017, and the best-performed statistics are indicated in bold type.

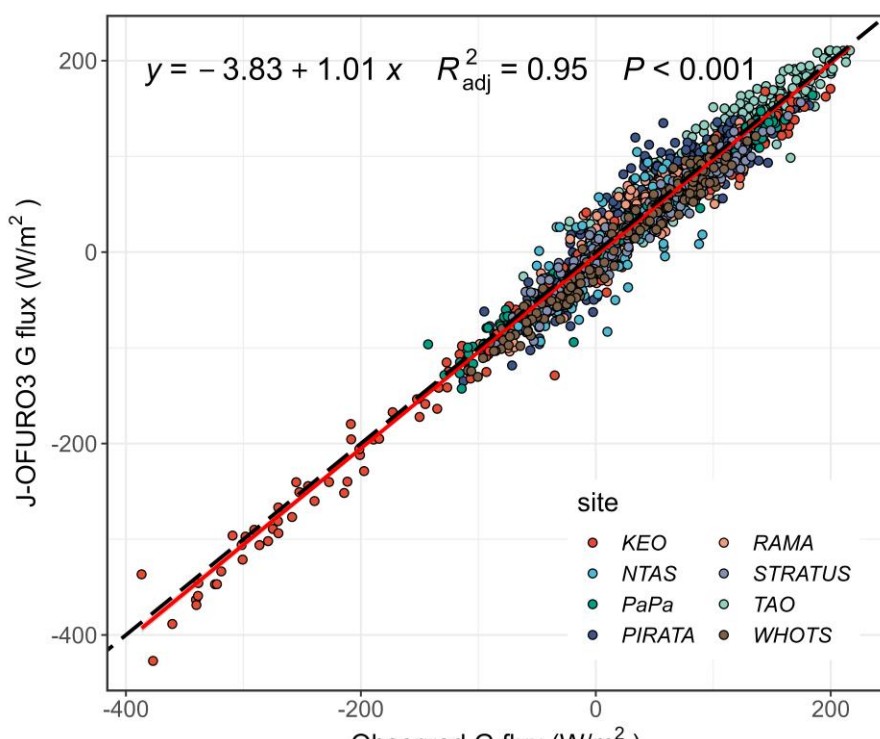

**Figure 5.** Assessment of heat storage (G) flux derived from remote sensed J-OFURO3 dataset against buoy observations. Distinct colors represent data collected from different buoy arrays.

**4.4 Estimating long-term global ocean surface heat fluxes by improved MEP model**

4.4.1 New estimate of global latent and sensible heat fluxes

After identifying the optimal driving dataset, this study employed the best-performed improved MEP method

(i, e., *M_0.24*, hereinafter referred to as MEP for simplicity, while the original MEP formula was denoted as

*MEP (ori)* for global scale estimation, producing new estimations of latent and sensible heat fluxes for the
period 1988-2017 (Table 5). The MEP model calculated the multi-year average $LE$ as 92.87 W·m$^{-2}$ and the
sensible heat flux as 12.27 W·m$^{-2}$ from 1988 to 2017. In comparison, $LE$ ranged from 88.95 (OAFlux) to
100.54 W·m$^{-2}$ (MERRA2), and $H$ ranged from 10.17 (J-OFURO3) to 13.16 W·m$^{-2}$ (MERRA2) for the other
four products. The original MEP method yielded estimates of $LE$ as 52.70 W·m$^{-2}$ and $H$ as 25.07 W·m$^{-2}$,
significantly underestimating $LE$ and overestimating $H$ compared to estimates from other products. As
previously demonstrated (Sections 4.1 & 4.2), the original MEP method overestimated $G$ (42.20 W·m$^{-2}$) and
exhibited notable deviations in the Bowen ratio. Therefore, the improved MEP method provided a more
reasonable global estimation of $LE$ and $H$.

Regarding the global spatial pattern (Fig.6), the MEP-derived latent heat exhibited higher values in low-

latitude regions but significantly decreased beyond 45° latitude. The highest $LE$ values were observed in the
southern Indian Ocean near Australia, the Pacific and Atlantic regions near South America, and the Indian
Ocean near southern Africa. The peak values were observed within western boundary current systems (ranged
from 200 to 260 W·m$^{-2}$), including the Gulf Stream in the North Atlantic and the Kuroshio in the western
North Pacific. Impacted by the variations in oceanic currents near the equator, two general areas of higher
$LE$ have emerged (Yu et al., 2011), leading to notably low $LE$ at the equator (88 W·m$^{-2}$), peaking at ~18°S at
132 W·m$^{-2}$ (Fig. 6 & Fig. 7). The MEP estimated $LE$ exhibited a similar spatial pattern with other four
products globally (Fig.6), particularly resembling OAFlux between 15°S and 15°N (Fig. 7). Overall, for the
region between 30°S and 30°N, the $LE$ values were ranked as follows: OAFlux < MEP < J-OFURO3 < ERA5
< MERRA2, which was consistent with the magnitude of available energy. For sensible heat, MEP-derived
H closely resembled that of ERA5 and MERRA2, with higher values predominantly occurred in two western
boundary current systems, the South Indian Ocean near Australia area, and the Arctic Ocean. The improved
MEP method mitigated the issue of overestimating $H$ in mid-to-high latitudes compared to its original form
(Fig.6l), resulting in a more realistic spatial pattern. In high latitudes, J-OFURO3 exhibited higher H values
than MEP and other comparable products in the Northern Hemisphere, with negative values observed
between 45°S and 55°S. MEP generally estimated $H$ within an intermediate range compared to other products,
displaying a distribution that was more reasonable than that of J-OFURO3 product.

**Table 5.** Global area-averaged multi-annual mean estimates of latent heat flux

| LE products | LE (W·m⁻²) | Evaporation (mm/yr) | H (W·m⁻²) | G (W·m⁻²) |
|---|---|---|---|---|
| MEP (0.24) | 92.8 | 1195.5 | 12.2 | 19.7 |
| ERA5 | 99.2 | 1277.8 | 12.0 | 34.2 |
| MERRA2 | 100.5 | 1294.3 | 13.2 | 35.5 |
| J-OFURO3 | 94.9 | 1222.2 | 10.1 | 19.7 |
| OAflux | 88.9 | 1145.1 | 10.4 | / |
| MEP (ori) | 52.7 | 678.5 | 25.1 | 42.2 |

Note: The period spans from 1988 to 2017. The MEP (0.24) denotes the improved MEP model, while MEP
(ori) represents the original MEP model.

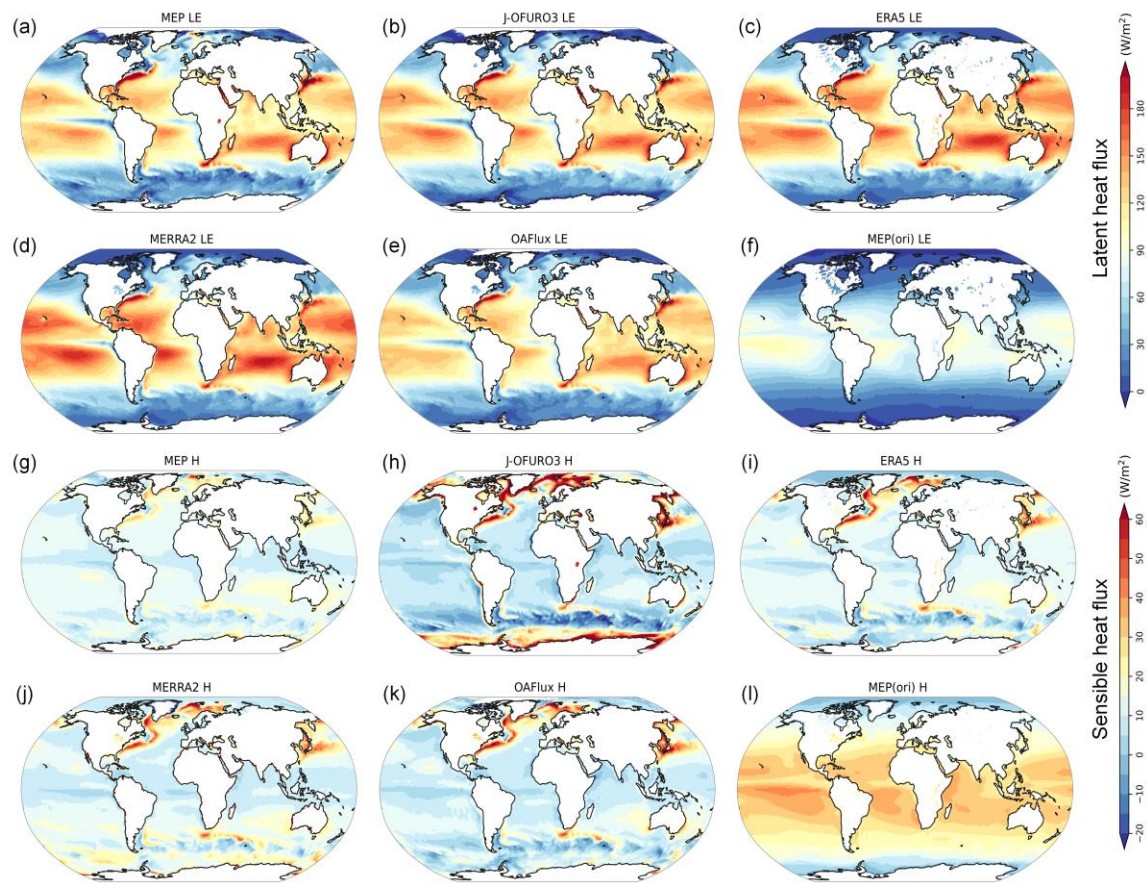


**Figure 6.** Global spatial maps of annual mean latent heat flux (*LE*) and sensible heat flux (*H*) during 1988-
2017. Panels (a)-(f) depict latent heat flux derived from the improved MEP method, J-OFURO3, ERA5,
MERRA2, OAFlux, and the original MEP method. Panels (g)-(l) show sensible heat flux from the same
datasets.

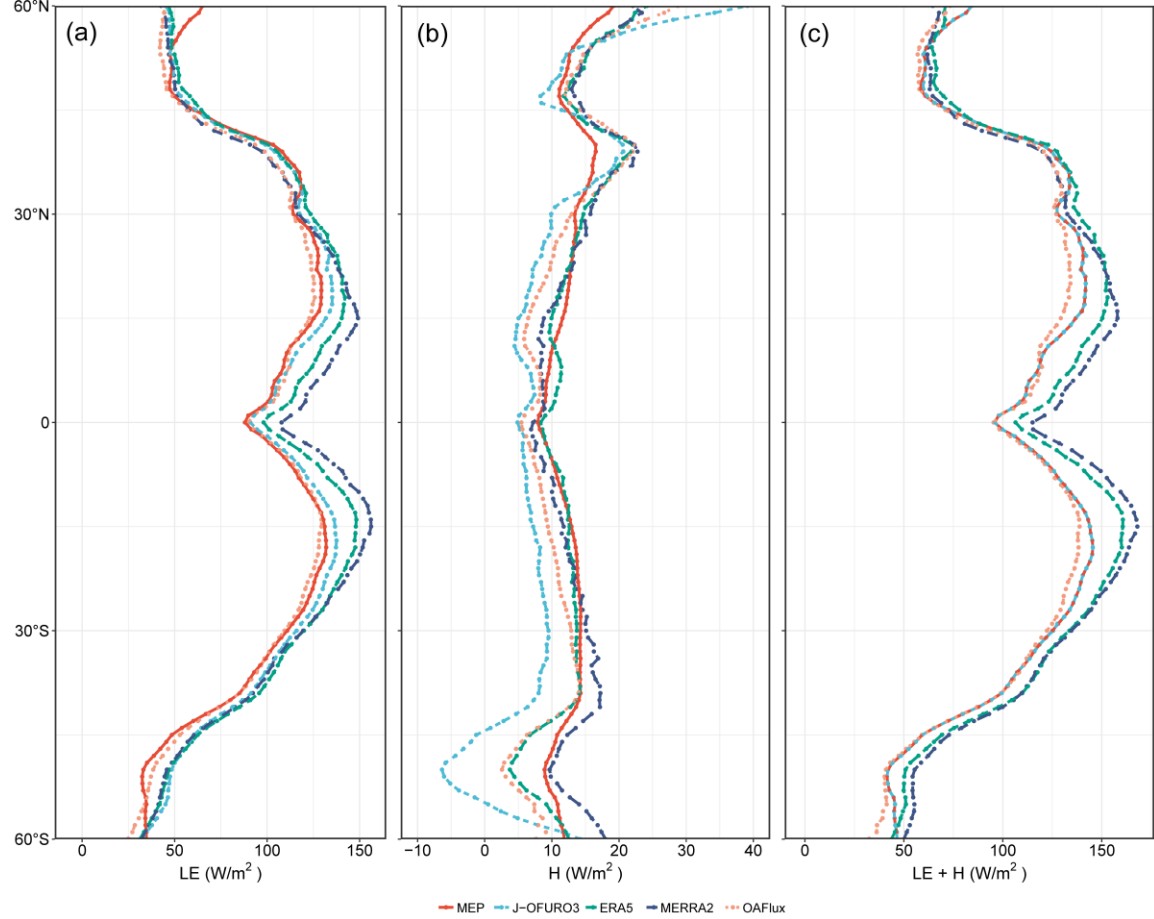


**Figure 7.** Meridional profiles of latent heat (left panel), sensible heat (middle panel) and their sum
representing available energy (right panel) for the period 1988-2017, produced by MEP, J-OFURO3, ERA5,
MERRA2, and OAFlux datasets.


### 4.4.2 Validation of global latent heat products against the observational sites

To evaluate the discrepancies between MEP estimated *LE* and other datasets, this study validated global-
scale *LE* at 129 observational sites (as depicted in Fig.8 & Table 6). MEP-estimated *LE* showed strong
consistency with buoy observations, achieving an $R^2$ of 0.79, a ME of 1.26 W·m$^{-2}$, and RMSE of 16 W·m$^{-2}$,
all surpassing those of alternative products, underscoring its superior performance. Moreover, the MEP
method exhibited superior performance with a higher NSE value of 0.77 and KGE of 0.89, demonstrating
enhanced accuracy, reliability, and robustness. According to the RMSE evaluation criterion, the ranking of
best-performed *LE* products was as: MEP, J-OFURO3, OAFlux, ERA5, MERRA2. In a recent
comprehensive assessment of 15 global ocean *LE* products (Tang et al., 2023), RMSE values ranged from
17.2 to 45.3 W·m⁻², in which J-OFURO3 emerged as the best-performing product with the lowest RMSE of
17.2 W·m⁻², highest correlation coefficient (R) of 0.89, and ME of 6.5 W·m⁻². Studies have also shown
minimal bias were given by J-OFURO3 on daily scale (Bentamy et al., 2017). This superior performance of
J-OFURO3 dataset can be attributed to the use of continuously updated bulk algorithms (COARE 3.0 version),
the ongoing optimization of near-surface parameters (Tomita & Kubota, 2018), as well as the improved
spatial resolution (0.25°). In this study, the improved MEP estimation of *LE* outperformed that of J-OFURO3,
demonstrating higher accuracy and lower error (ME=1.26 W·m⁻²), thereby establishing it as the most accurate
global *LE* product currently available.

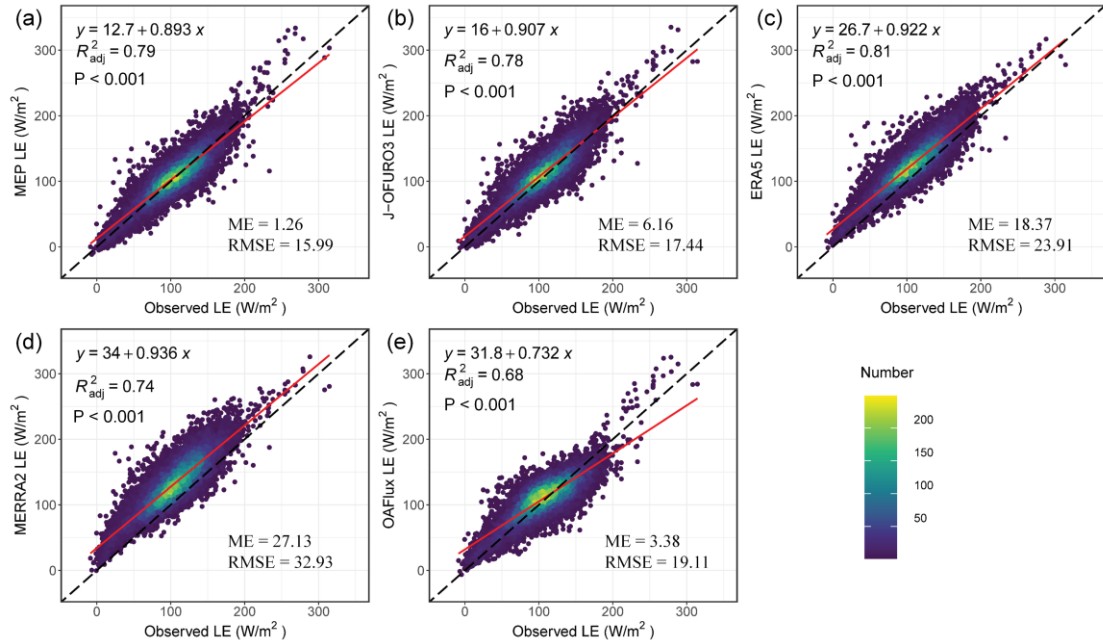


**Figure 8.** Scatter density plots of latent heat flux taken from different products versus observations from 129
buoy stations during the period 1988-2017: (a) Improved MEP model, (b) J-OFURO3, (c) ERA5, (d)
MERRA2, and (e) OAFlux. A total of 15444 records of latent heat observations are included.

**Table 6.** Evaluation of latent heat flux from different methods against buoy observations

| Products | $R^2$ | ME (W·m⁻²) | MAE (W·m⁻²) | RMSE (W·m⁻²) | PBIAS (%) | NSE | KGE |
|---|---|---|---|---|---|---|---|
| MEP | 0.79 | **1.3** | **12.2** | **15.9** | **1.2** | **0.77** | **0.89** |
| J-OFURO3 | 0.78 | 6.3 | 13.4 | 17.4 | 5.8 | 0.73 | 0.87 |
| ERA5 | **0.81** | 18.4 | 19.9 | 23.9 | 17.3 | 0.48 | 0.80 |
| MERRA2 | 0.74 | 27.1 | 28.1 | 32.9 | 25.5 | 0.02 | 0.70 |
| OAFlux | 0.68 | 3.4 | 14.9 | 19.1 | 3.2 | 0.67 | 0.79 |

Note: The evaluation period spans from 1988 to 2017, and the best-performed statistics are indicated in bold
type.

 4.4.3 Comparisons of Bowen ratios

The improved MEP model achieved accurate *LE* estimation after refining the process of partitioning the
surface energy budget, specifically through revisions to the Bowen ratio. The improved MEP method notably
decreased the global-scale Bowen ratio, as illustrated in Fig. 9 and 10. Regarding latitude averages, the
Bowen ratio of the original MEP formula ranged from 0.37 to 1.48 (with a median of 0.80), whereas the
modified MEP Bowen ratio ranged from 0.09 to 0.35 (median of 0.18). Specifically, in the low-latitude region
(10°S-10°N), the Bowen ratio of the modified MEP formula decreased from 0.37 to approximately 0.1,
aligning closely with the Bowen ratios obtained from other reanalysis products (MERRA2, ERA5, OAFlux,
and J-OFURO3). Globally, the median Bowen ratios of the products were as follows: MERRA2 (0.15), MEP
(0.12), ERA5 (0.09), OAFlux (0.08), and J-OFURO3 (0.06). Spatially, the MEP Bowen ratio resembled
ERA5 in mid to low latitudes but exhibited deviations from other products at high latitudes, where those
products showed fluctuating changes in the Bowen ratio (Fig.10). For instance, other products displayed
abrupt transitions from negative to positive Bowen ratios in the Arctic and Antarctic regions, whereas MEP-
derived values demonstrated greater stability in variations at higher latitudes. This discrepancy was likely
due to the reanalysis products relying on the bulk method, which was sensitive to variations in wind speed
and temperature gradients, leading to errors in simulating high wind speeds at the poles and causing
fluctuations in latent and sensible heat. In contrast, the MEP model strictly adheres to energy conservation
principles and operates independently of wind speed and temperature gradients, resulting in a more accurate
estimate of the Bowen ratio. For example (Fig.S9), at the high-latitude PAPA buoy site (144.9°W, 50.1°N),
the Bowen ratio estimated by MEP (median 0.24) closely matched the observed Bowen ratio (median 0.23).
In contrast, all the other products underestimated the Bowen ratio, with J-OFURO3 (median -0.09) and
OAFlux frequently exhibiting negative values. The Bowen ratio derived from MEP fit well with a
Generalized Additive Model (GAM) (Fig.9). The implicit functional relationship between Bowen ratio and
latitude was expressed as ($R^2 = 0.996$, $p < 0.001$): $B_{oa}\ (lat) = 0.207218 + f(lat) + \varepsilon$, where *f(lat)* represents a
smoothing function derived from a smooth curve, and $\varepsilon$ denotes the error term. However, the specific
functional form of *f(lat)* cannot be explicitly determined. Therefore, a polynomial regression method was
employed to explicitly fit $B_{oa}$ and *lat*, resulting in ($R^2 = 0.91$, $p < 0.001$): $B_{oa} = 9.97 \times 10^{-2} - 3.45 \times 10^{-4} \times lat$
$+ 4.71 \times 10^{-5} \times lat^2 + \varepsilon$ (as in Fig.S10). This equation can serve as a reference for partitioning surface energy
in data-sparse oceanic regions.

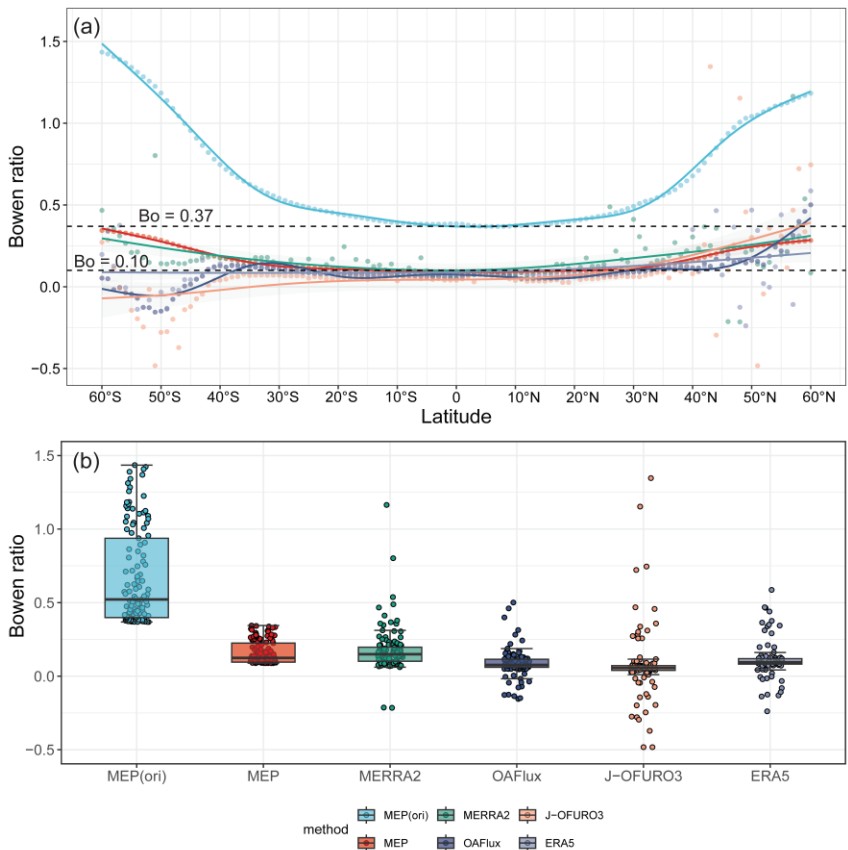


**Figure 9.** Global ocean latitudinal averaged Bowen ratio derived by the MEP method and four other products
from 1988 to 2017. (a) Latitudinal averaged Bowen ratio derived from the MEP model using original and
modified Bowen ratio formulas, with points fitted by a generalized additive model (GAM). (b) Statistical
distribution of the latitudinal annual mean Bowen ratio.

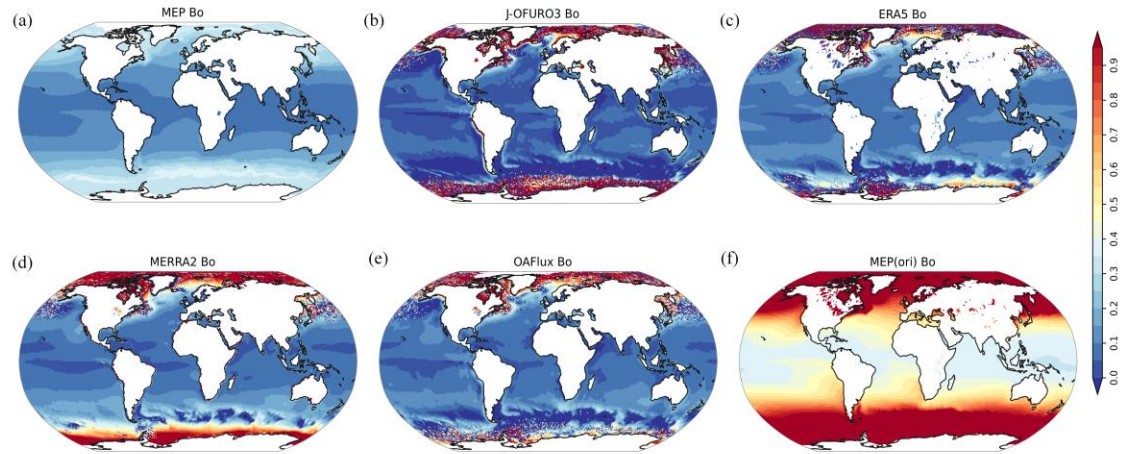


**Figure 10.** Global distribution of ocean annual mean Bowen ratio during 1988-2017: (a) Improved MEP
method, (b) J-OFURO3, (c) ERA5, (d) MERRA2, (e) OAFlux, and (f) MEP original method.

## 4.5 Spatial-temporal variability of ocean evaporation

Heat flux reflects the energy exchange between the ocean and the atmosphere, while evaporation (*ET*) reflects moisture exchange within the water cycle. The spatiotemporal patterns of evaporation were analyzed using the Sen's slope and Mann-Kendall test methods (Fig. 11). For the global ocean, approximately 74% of the regions showed an increasing trend, with about 27% of the grids exhibiting statistically significant increases ($p < 0.05$). In contrast, 26% displayed a decreasing trend, with only 5% of the grids showing statistically significant decrease ($p < 0.05$). For the whole periods, the regions with the highest increasing trends were predominantly observed near western boundary current systems, the convergence zones of the East Australian Current and the South Equatorial Current, and the convergence zones of the Eastern South Equatorial Current and the Brazil Current along South America. Decreasing trends were primarily observed in equatorial regions of the Pacific and Atlantic Oceans, as well as near the Labrador and Kuroshio currents and north of the Antarctic Circle. It was indicated that regions with significant increases (decreases) in evaporation generally correspond closely to the distribution of major warm currents (cold currents) spatially. However, global ocean evaporation experienced a notable shift around 2003, as illustrated in Figs.11b and 11c. The downward trend observed from 2003 to 2017 counteracted a significant portion of the growth trend that occurred during the previous 16 years (1988-2003), particularly evident in the mid-latitude regions (15ºS-20ºN). In the middle-to-low latitudes (0º-30ºN), nearly all ocean grids exhibited opposite trends around 2003. Spatially, regions that displayed the largest increasing trends during 1988-2003 transitioned to show the most substantial decreasing trends between 2003 and 2017. This includes regions associated with western boundary current systems, convergence zones of the East Australian Current and the South Equatorial Current, as well as equatorial regions of the Pacific and Atlantic Oceans (Fig. 11c). To further investigate the shift in ocean evaporation after 2003, we analyzed the interannual variability of global annual mean area-weighted evaporation using all available datasets (as shown in Fig.12).

Over the multi-year period from 1988 and 2017, MEP, J-OFURO3, ERA5, and MERRA2 all exhibited significant increasing trends in *ET*. MEP estimated an evaporation increase rate of 2.31 mm/year, whereas OAFlux showed a non-significant trend (Fig. 12). While different datasets revealed varying magnitudes of evaporation changes, most exhibited a similar temporal pattern: an increasing trend from 1988 to around 2003, followed by a hiatus during 2003-2010, and ultimately a decreasing trend after 2010 (Fig. 12a).

Specifically, MEP indicated an increasing trend in evaporation of 3.58 mm/year from 1988 to 2010, followed
by a decrease of 2.18 mm/year after 2010 (Fig. 12a). The slowdown and transition of evaporation during
2003-2010 aligned with the concept of a "global warming hiatus" (Medhaug et al., 2017; Sung et al., 2023),
referring to the period when global mean surface air temperatures did not continue to rise between 1988 and
2012. Previous studies have proposed four potential explanations for this global warming hiatus: internal
variability, external drivers, the Earth's response to $CO_2$, and radiative forcing (Medhaug et al., 2017). This
study indicates that changes in radiative forcing (Fig. 12b) can significantly affect the interannual variability
of evaporation (Fig. 12a) and surface temperature (Fig. 12c). This finding is consistent with previous research
that attributed more than 50% of the uncertainty in MEP-modeled fluxes to the radiation term (Huang et al.,
2017). Although surface temperature began to increase after 2012, the decrease in available energy remained
the primary driver behind the decline in evaporation.

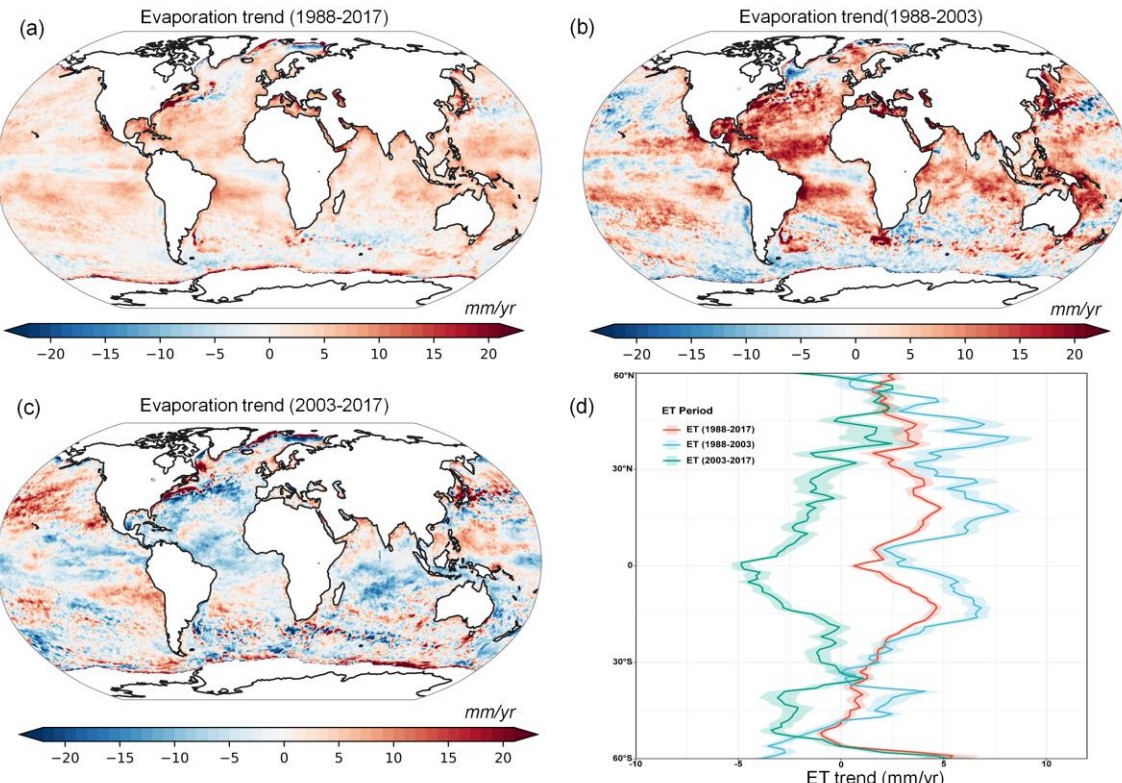


**Figure 11.** Spatial distribution of multi-year trends in ocean evaporation estimated by the improved MEP
method during (a) the period 1988-2017, (b) the period 1988-2003, (c) the period 2003-2017, and (d) the
latitudinal average changes across three different periods.

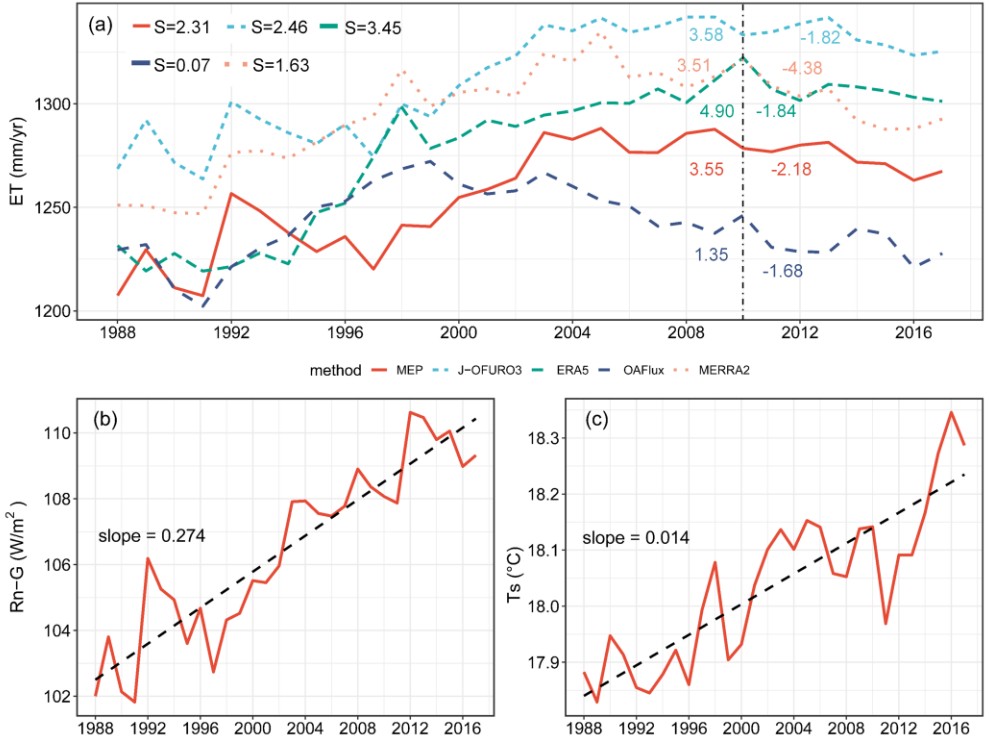


**Figure 12.** Time series of area-averaged multi-annual mean evaporation from the improved MEP method (a),
available energy (b), and sea surface temperature (c) over the global oceans during 1988-2017. The black
dotted line in panel (a) marks the year 2010, and the label "S = 2.31" indicates that the MEP-estimated global
multi-annual mean evaporation increased at a rate of 2.31 mm/year during 1988-2017, with change rates of
different ET datasets represented by various colors. The black dashed lines in panels (b) and (c) denote the
linear regression lines.
**5. Discussion**
**5.1 Quantifying the impact of heat storage and radiation with sensitivity analysis**
The sensitivity analysis revealed the significant influence of input variables on latent heat flux derived from
the MEP model. Notably, the heat storage ($G$) exhibited seasonal variations with both positive and negative
values (Fig. 13). Positive $G$ values coincided predominantly with summer in the Northern Hemisphere
(winter in the Southern Hemisphere), specifically from June to August (Fig. 4 and Fig. S5). During this
season, intensified solar radiation enhances the net energy input ($R_n$) at the ocean surface, leading to heat
absorption and retention. Consequently, the energy available ($R_n$ - $G$) for evaporation diminishes. The
analysis indicated that $R_n$ significantly influenced the energy-driven evaporation process, with a sensitivity
coefficient exceeding 1 (median 1.74), highlighting its pivotal role. In contrast, $G$ negatively impacted
evaporation, as indicated by a sensitivity coefficient of -0.74. Specific humidity (median 0.08) and sea surface
temperature had relatively minor effects, consistent with previous MEP model findings focused on terrestrial
surfaces (Isabelle et al., 2021).
Conversely, negative values of heat storage predominate during winter, particularly from December to
February in the Northern Hemisphere (June to August in the Southern Hemisphere). Despite reduced solar
radiation during this period, residual heat stored from summer gradually releases into the atmosphere,
resulting in greater energy output than input. This surplus energy increases the available energy for
evaporation, leading to a positive sensitivity coefficient for $G$ (median 0.29), second only to $R_n$ (median 0.71).
Consequently, this process generally reduces sea surface temperature, resulting in a negative sensitivity
coefficient for surface temperature. Overall, these findings underscored the significant influence of $R_n$ on
latent heat flux, with $G$ ranking as the second most influential variable in MEP estimates over ocean surfaces.
For instance, a 10% decrease in positive $G$ yielded a 7.4% increase in evaporation, while a 10% increase in
negative $G$ resulted in a 2.9% increase in evaporation, assuming other variables remain constant. Thus, $R_n$
and $G$ emerged as two primary drivers of oceanic evaporation, with humidity and temperature exerting
minimal influence.
The accuracy of available energy estimates significantly $LE$ simulations, as it serves as the direct energy
source for partitioning latent and sensible heat fluxes. Although the bulk methods (e.g., COARE 3.0
algorithms) used for estimating heat fluxes are independent of surface energy budget allocation, discrepancies
in $LE$ estimation still correlate strongly with validated biases against observations in available energy
estimates (see Tables 3 and 4). Notably, the MERRA2 product exhibited higher errors in simulating $R_n$ and
$G$ compared to observations, leading to significant biases in $LE$ estimation. In contrast, the ERA5 product
demonstrated superior performance in simulating $R_n$ and $G$, thereby achieving higher accuracy in $LE$
estimation. Consequently, the energy-balance-based MEP model excels in accurately estimating surface heat
fluxes by directly reflecting energy allocation. Unlike bulk methods, the MEP approach reduces sensitivity
to temperature and humidity gradients, thereby minimizing uncertainties in $LE$ simulations (Pelletier et al.,
2018). This advancement enhances the MEP model's utility in global energy and water cycle research,
particularly pertinent for future climate change studies.

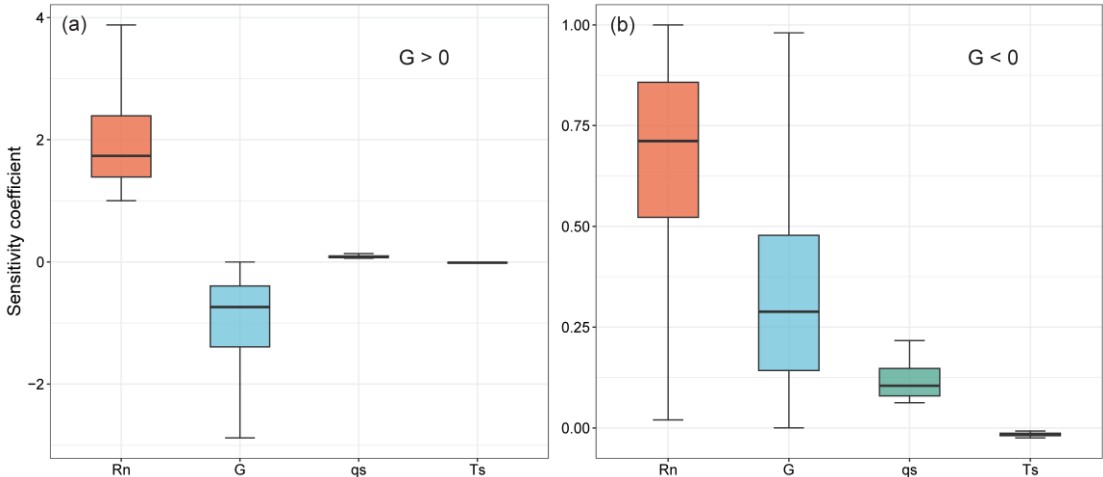


**Figure 13.** Sensitivity coefficient associated with inputs variables for the improved MEP method at all 129
buoy stations: (a) for positive $G$ values, and (b) for negative $G$ values.


### 5.2 Discrepancy of empirical Bowen ratio formulas

$B_o$ plays a crucial role in understanding the surface energy partitioning process. In this study, four empirical

formulas were utilized to modify the MEP model and evaluated against the observed $LE$, each with distinct

conditions of applicability and suitability for integration with the MEP model: (1) $B_{oa}=0.63B_o{}^*-0.15$ was

derived from direct observational data fitting (Hicks and Hess, 1977). This formula was applicable for surface

temperatures above 16°C, particularly within latitudes between 40°N and 40°S, making it more suitable for

lower latitude regions; (2) $B_{oa}=0.79B_o{}^*-0.21$ was derived using the Priestley–Taylor model under advection-

free conditions (Priestley and Taylor, 1972). The coefficients were determined based on a mean α value of

1.26, although this value can vary in practice. Recent studies have revealed significant discrepancies due to

the neglect of interaction between variations in $R_n$ and $T_s$ (Yang & Roderick, 2019). To address this limitation,

the equation (3) $B_{oa}=0.24B_o{}^*$ was developed based on the maximum evaporation theory, considering the

feedback mechanisms between $R_n$ and $T_s$ while assuming that $G$ is small or negligible. The empirical

coefficient 0.24 was determined by fitting $B$ and $T_s$ across the global ocean surface (Yang & Roderick, 2019);

and (4) $B_{oa}=0.37B_o{}^*-0.05$ was formulated based on principles derived from atmospheric boundary layer (ABL)

theory (Liu & Yang, 2021), with coefficients also fitted from relationships between $B_{oa}$ and $T_s$. It should be

noted that the derivations of $B_{oa}=0.24B_o{}^*$ and $B_{oa}=0.37B_o{}^*-0.05$ were based on fitting using $LE$ from the

OAFlux dataset rather than direct buoy observations. Overall, the MEP model incorporated with $B_{oa}=0.24B_o{}^*$,

exhibited superior accuracy at both localized and global scales, effectively mitigating the underestimation of
*LE* in its original estimates.
### 5.3 Contributions and implications of this study
The main contributions of this study include: (1) The MEP model's energy balance equation over water
surfaces was revised to explicitly consider heat storage effect. This correction highlights the importance of
heat storage in estimating *LE*. (2) The energy partitioning of the MEP model was revised to incorporate
empirical Bowen ratio formulas, significantly improving the heat flux estimations. (3) This study conducted
the first thorough global assessment of heat storage using extensive buoy observations and remote sensed
data, enabling the MEP model to produce the most accurate global *LE* estimates. This study addresses the
issue of underestimating *LE* by the original MEP model, increasing the global average *LE* from 53 $W \cdot m^{-2}$ to
93 $W \cdot m^{-2}$, while reducing sensible heat flux from 25 $W \cdot m^{-2}$ to 12 $W \cdot m^{-2}$, improving the partitioning of energy
budget. The improved MEP model provided precise *LE* estimates compared to existing datasets like J-
OFURO3, ERA5, MERRA2, and OAFlux, enabling it to become a valuable benchmark dataset for global
evaporation studies.
From a methodological perspective, the improved MEP method emerged as a novel approach for
estimating energy fluxes that diverges from traditional bulk methods. The conventional bulk method requires
multiple input parameters, including air temperature, specific humidity, wind speed, sea surface temperature,
atmospheric pressure, and the observational height of all parameters (Fairall et al., 2003; Tomita et al., 2021).
This method demands numerous input variables, and the estimated fluxes are highly sensitive to changes in
temperature and humidity gradients. In contrast, the improved MEP model requires only net radiation, heat
storage, surface temperature, and atmospheric pressure to simultaneously obtain latent and sensible heat
fluxes, making it more flexible to operate and robust against variations in input variables. Furthermore, the
improved MEP model is not constrained by the magnitude of heat storage and theoretically can be applied
across various temporal scales (including sub-daily and daily), beyond the monthly scale used in this study.
This underscores the applicability of the MEP method in addressing the constraints of traditional bulk
methods, providing another independent approach to estimating heat fluxes across diverse environmental
conditions.
This study applied the improved MEP model to ocean surface, with potential for future extension to lake
and reservoir surfaces. Compared to the Penman model for water body evaporation (Tian et al., 2022; Zhao
et al., 2022; Bai et al., 2023), the major advantage of MEP method lies in its independence from wind speed,
provided that heat storage can be determined using an equilibrium temperature-based approach (McMahon
et al., 2013; Zhao & Gao, 2019). The global *LE* dataset generated in this study, due to MEP's insensitivity to
variations in air temperature and humidity, can be applied in research related to ocean salinity (Liu et al.,
2019), ocean warming (Cheng et al., 2022), and global climate change and water cycle studies (Konapala et
al., 2020).
## 5.4 Limitations
The improved MEP method proposed in this study offers a novel approach for estimating ocean heat fluxes,
producing a validated long-term global dataset with high accuracy and spatiotemporal continuity. Despite its
advancements, the proposed MEP method has several limitations that require further refinement: (1)
Uncertainty of Driving data: The input variables of net radiation, heat storage, and sea surface temperature
for the MEP model were sourced from the state-of-the-art satellite-based J-OFURO3 dataset. This dataset
was constructed using observations from multiple satellite sensors. The net radiation in J-OFURO3 was
derived by combining data from the CERES and the International Satellite Cloud Climatology Project
(ISCCP) via the creeping sea fill method, along with twelve global sea surface temperature products (Tomitta,
2019). Consequently, the uncertainty of the MEP-estimated fluxes may arise from biases in input data derived
from various satellite sensors and their associated analysis methods. Therefore, it is essential to integrate
multiple approaches to assess the uncertainty associated with the input datasets. Moreover, due to the limited
temporal duration of the J-OFURO3 dataset, future work should utilize input datasets with longer time series,
finer spatio-temporal resolution (Liang et al., 2022), and higher accuracy to advance ocean heat flux
estimations using the MEP method. (2) Heat Storage Determination: This study did not employ a direct
calculation method to obtain heat storage. Given the unclear relationship between heat storage and changes
in ocean heat content at varying depths (as shown in Table 4), we utilized an energy balance residual-based
approach to indirectly estimate heat storage. Consequently, this may render the MEP method susceptible to
uncertainties in heat storage data derived from auxiliary flux datasets. Future research should focus on
understanding the relationship between ocean heat content changes in the upper 100m and heat storage, with
the goal of establishing a functional relationship between water column temperature at different depths and
heat storage. (3) Bowen Ratio Improvement: Accurate determination of the Bowen ratio in high-latitude
regions remains challenging. The Bowen ratio derived from the MEP method showed significant
discrepancies with other datasets in these areas (Fig. 10), particularly in sea-ice-covered Arctic regions, where
other datasets exhibited notable overestimations and irregular fluctuations. Therefore, incorporating more
observational data from high-latitude regions is essential for a better understanding of energy partitioning
patterns.

## 6. Data availability


The GOHF-MEP dataset produced by the MEP method, which includes global latent heat flux and sensible
heat flux at a monthly scale from 1988 to 2017, can be freely downloaded from the Figshare platform
(https://doi.org/10.6084/m9.figshare.26861767.v2, Yang et al., 2024). All the datasets used in this study are
publicly available online and are described in the "Data Materials" section.

## 7. Conclusions


In this study, we developed a new global ocean heat flux product (GOHF-MEP) covering the period from
1988 to 2017. This product is grounded in a maximum entropy production theory framework, incorporating
heat storage impacts and Bowen ratio adjustments. GOHF-MEP represents the first energy-balance-based
dataset that diverges from existing global ocean heat flux datasets derived from bulk methods. To assess the
accuracy of the input variables for the maximum entropy production framework, we utilized five global
datasets, including two remote sensing-based and three from reanalysis-based, along with four global datasets
of heat storage derived from the energy balance equation and ocean heat content changes. We employed data
fusion methods, including arithmetic averaging and the Bayesian three-cornered hat method, to identify
optimal input datasets through validation against observations. The performance of the newly produced
GOHF-MEP dataset was evaluated against extensive observations from 129 globally distributed buoy
stations using multiple statistical metrics. It was also compared with four auxiliary products: J-OFURO3,
ERA5, MERRA2, and OAFlux. Moreover, we analyzed the long-term spatial-temporal variability of ocean
latent heat flux. Ultimately, we investigated the impacts of ocean heat storage, net radiation, and Bowen ratio
changes on heat flux estimations and surface energy partitioning.

The MEP framework provides new estimates of global heat fluxes. The MEP-estimated long-term

annual mean latent heat flux is 93 $W \cdot m^{-2}$ (equivalent to 1196 mm/year of evaporation) during the period from
1988 to 2017. This estimate is intermediate compared to other global flux products, which range from 90
$W \cdot m^{-2}$ (OAFlux) to 101 $W \cdot m^{-2}$ (MERRA2). The MEP-estimated sensible heat flux is 12 $W \cdot m^{-2}$, falling within
the range of 10.17 $W \cdot m^2$ (J-OFURO3) to 13 $W \cdot m^2$ (MERRA2) reported by other current products. Compared
with previous heat flux products, the MEP-estimated latent heat demonstrated higher accuracy when
validated against observations, with a ME of 1.26 $W \cdot m^{-2}$, a RMSE of 16 $W \cdot m^{-2}$, and a KGE value of 0.89,
outperforming all other contemporary global products. Approximately 74% of oceanic regions experienced
an increasing trend in evaporation from 1988 to 2017. In terms of long-term temporal variability, the global
annual mean evaporation exhibited an increase rate of 3.58 mm/year from 1988 to 2010 but subsequently
declined at a rate of 2.18 mm/year from 2010 to 2017, which was consistent with changes in surface available
energy.

This study demonstrates that the improved MEP framework has significantly improved the accuracy of

the original MEP theory, addressing both the underestimation of latent heat and the overestimation of sensible
heat flux. This improvement was achieved by incorporating the impact of heat storage and modifying the
Bowen ratio formula within the MEP theory. The consideration of heat storage resolved the issue of seasonal
phase mismatches (approximately 6-month lags) between MEP estimates and buoy observations. Building
upon this improvement, this study further optimized the energy partitioning process by correcting the Bowen
ratio, linearly adjusting the equilibrium Bowen ratio to align with actual conditions. Four empirical Bowen
ratio formulas for modifying the MEP method were assessed globally, identifying $B_{oa}=0.24B_o^*$ as the most
accurate formula for estimating latent heat flux within MEP method. The impact of heat storage on estimating
heat fluxes was quantified through sensitivity analysis. Net radiation and heat storage were identified as the
primary drivers of evaporation estimates. A 10% decrease in positive heat storage led to a 7.4% increase in
evaporation, whereas a 10% increase in negative heat storage resulted in a 2.9% increase.

Compared to existing bulk methods, the MEP model offers several advantages, including the

requirement for fewer input variables, independence from wind speed, and insensitivity to variations in
temperature and humidity. The MEP-derived ocean heat flux dataset has been validated and provides accurate
estimates of latent heat flux. Additionally, this MEP method can be applied to estimate evaporation from
other deep-water surfaces, such as lakes and reservoirs where heat storage is significant. Overall, the MEP-
derived ocean heat flux dataset provides high global accuracy, fine spatial resolution (0.25°), and extensive
long-term temporal records. This dataset is expected to be valuable for applications related to global ocean
warming, hydrological cycles, and their interactions with other Earth system components in the context of
climate change.
**Supplement.** The Supplementary material related to this article is submitted during submission.

**Author contributions.** YY and HS developed the methodology and designed the experiments, WZ
contributed the conceptual design, YY and WZ collected and processed the data, YY wrote the first draft of
the paper under the supervision of other authors. All authors participated in the revising and editing of the
manuscript.

**Competing interests.** The contact author has declared that none of the authors has any competing interests.

**Acknowledgments.** We acknowledge the GTMBA Project Office of NOAA/PMEL for providing the Global
Tropical Moored Buoy Array observations. This study was primarily funded by the Third Xinjiang Scientific
Expedition Program (Grant No.2022xjkk0105) (H.S.). The authors acknowledge funding from the NSFC
project (52079055, 52011530128). H.S. and W.Z. acknowledges funding from the NSFC-STINT projects
(No. 202100-3211 and CH2019-8281). W.Z. was supported by the grants from Swedish Research Council
VR (2020-05338), and Swedish National space Agency (209/19).

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
