# Peer review of "storage and Bowen ratio optimizations"

_Earth System Science Data, 2024_

## Author Comment (AC1)

**Responses to reviewers – ESSD-2024-420**

**"Global ocean surface heat fluxes revisited: A new dataset from maximum entropy production framework with heat storage and Bowen ratio optimizations"**, by Yong Yang, Huaiwei Sun, Jingfeng Wang, Wenxin Zhang, Gang Zhao, Weiguang Wang, Lei Cheng, Lu Chen, Hui Qin, Zhanzhang Cai, submitted to ***Earth System Science Data***.

Dear Editors and Reviewers,

Thank you for your letter and for the reviewers' comments concerning our manuscript **ESSD-2024-420**. We appreciate editors and reviewers very much for the positive and constructive comments and suggestions.

Those comments are all valuable and very helpful for revising and improving our paper, as well as providing the important guidance for our researches. We have addressed all the comments carefully and made the revisions which we hope will meet your approval.

Below are our point-by-point responses to the comments raised by the referee. Comments of reviewers are marked with the blue color, and the corresponding responses are in black (begins with bold words of **"Response"**). Corresponding changes in the text of the manuscript appear in red font.

**Responses to Reviewer #1**

Reviewer #1: General Comments:
* * *
This research presents a new global ocean heat flux dataset developed using an innovative energy-balance-based method. The authors introduce a new approach grounded in energy allocation principles, utilizing modified Maximum Entropy Production (MEP) theory to estimate oceanic heat fluxes. The methodology is robust, the calculation process is well-structured, and the dataset is in good accuracy. The paper is well written, and the statistical and geographical analyses are conducted appropriately. The manuscript falls within the scope of the ESSD journal. Minor revisions need to be considered before publication. Please find details below:

**Response:** We appreciate reviewer's positive comments, and we will further improve our manuscript by fully address the reviewer's comments in the following.

Specific comments:

Title and Abstract:

(1) This paper develops a new global ocean heat flux dataset using the MEP framework, incorporating heat storage and Bowen ratio optimizations. I recommend the author designate a representative name for this dataset, such as "Maximum Entropy Production-based Global Ocean Heat Flux (OHF-MEP)."

**Response:** Thanks very much for your suggestion, which we strongly agree with. We decide to use the abbreviation (GOHF-MEP) to represent our dataset throughout the text.

"The 0.25° monthly global ocean heat flux dataset based on the Maximum Entropy Production method (GOHF-MEP) for 1988–2017, is publicly accessible at…"
(Line 44-45, in "1. Introduction" section)

"The GOHF-MEP dataset produced by the MEP method…"

(Line 801, in "6. Data availability" section)

"In this study, we developed a new global ocean heat flux product (GOHF-MEP) covering the period from 1988 to 2017"

(Line 806, in "7. Conclusions" section)

"The performance of the newly produced GOHF-MEP dataset was evaluated against extensive observations…"

(Line 814-815, in "7. Conclusions" section)

(2) Line 25, The author should provide a definition of the Bowen ratio upon its first mention to ensure clarity.

**Response:** Thank you for your comment. The corresponding revisions are as follows:

"This study derived global ocean heat fluxes using the MEP theory, incorporating the effects of heat storage and adjustments to the Bowen ratio (the ratio of sensible heat to latent heat)."

(Line 27-28, in "Abstract" section)

"$B_o$ is crucial for understanding the global ocean energy partitioning process (Hicks & Hess, 1977)."

(Line 100, in "1. Introduction" section)

(3) Line 27-29, Consider rewording for better clarity. It should be "enhance/improve the model performance"

**Response:** Thank you for your helpful comment. The corresponding revisions are as

follows:

"The model was first evaluated using observed data from buoy stations, and the Bowen ratio formula that most effectively enhances the model performance was identified."
(Line 30-31, in "Abstract" section)

(4) Lin 29-31, The statement needs to be re-organized for better comprehension. Specifically, accounting for heat storage and adjusting the Bowen ratio were conducted within the MEP model or after the calculations?

**Response:** Thanks for your helpful comment. We have re-organized this sentence to improve the clarity. Accounting for heat storage and adjusting the Bowen ratio were conducted within the MEP model, and the corresponding revision is as:

"By incorporating the heat storage effect and adjusting the Bowen ratio within the MEP model, the accuracy of the estimated heat fluxes was significantly improved, achieving an $R^2$ of 0.99 (regression slope: 0.97) and a root mean squared error (RMSE) of 4.7 $W·m^{-2}$ compared to observations"
(Line 32-34, in "Abstract" section)

**Introduction:**
(5) This section is well written and organized, it presents the significance of ocean evaporation estimation, the limitations of current bulk methods, the introduction of the MEP method, and the ways to improve the model estimation through Bowen ratio fitting. However, a brief overview of existing ocean heat flux datasets (such as algorithms and accuracy) would help clarify the necessity for developing a new dataset.

**Response:** We appreciate for your valuable suggestion. We have added a brief overview of the current accessible global ocean datasets, including the classification, resolution, algorithms and representative products (in the last paragraph of the

Introduction). Also, the necessity for developing a new dataset with a new method and fine spatial resolution is clarified. Moreover, we have included a detailed description on the current ocean heat flux datasets in the section of "3.3 Global turbulent heat flux datasets for evaluations", including the algorithms, resolution, and variables (Table 2). The corresponding revisions are given as follows:

"Current global ocean surface heat flux datasets can be classified into five categories based on their deriving approaches (Tang et al., 2023): remote sensing-based (e.g., J-OFURO3), atmospheric reanalysis-based (e.g., ERA5), machine learning-based (e.g., OHFv2), in-situ based (e.g., NOC), and hybrid-based (e.g., OAFlux) approaches. Compared to terrestrial flux products, these ocean flux products generally have a coarser spatial resolution ranging from 0.25º to 1.875º. Recent studies have conducted comprehensive assessments of global ocean heat flux datasets regarding their accuracy and error characteristics across spatial and temporal scales (Bentamy et al., 2017; Tang et al., 2023). However, substantial discrepancies remain among these datasets, particularly in terms of spatial patterns, annual means, and interannual variabilities. Therefore, developing a new global dataset using the innovative method could advance our understanding of deriving algorithms, improve temporal and spatial coverage of flux variables with a higher accuracy, and provide alternative reference to assess ocean surface heat fluxes in various applications."
(Line 117-127, in "Introduction" Section)

(6) Line 49, "A key component of this regulation is ocean evaporation (latent heat)", "latent heat" is repeated and not necessary.

**Response:** Thank you for pointing this out. The revised text reads as follows:

"A key component in this system is ocean evaporation, which accounts for approximately 86% of atmospheric water vapor, being the primary driver of the global hydrological cycle (Yu, 2011)."

(Line 50-52, in "Introduction" Section)

**Methods:**

(7) Line 177 and Line 194, Remove the space before "where".

**Response:** Thank you for pointing this out. We have revised it and the text reads as follows:

"where $B_o^*$ is the equilibrium Bowen ratio, which denotes the theoretical ratio of…"
(Line 194)

"where $B(\sigma)_{a1} \sim B(\sigma)_{a4}$ represent the four empirical Bowen ratio formulas for comparisons in this study."
 (Line 211)

(8) Line 195, Consider rewording for better clarity. "Thus, the improved MEP model is complemented as…".

**Response:** Thanks for your careful checks. It is a writing error here, and we have revised it as:

"Thus, the workflow of the improved MEP model was conducted as:"
(Line 212)

(9) Line 309, "…at different depths with the observed G (derived as Rn-LE-H) (Fig.S1)", Should it be Table.S1? check and make sure it.

Response: Thank you for correcting this writing error. It should be "Table S1" rather than "Fig.S1", and the revision is as:

*"…this study compared the OHC changes at different depths with the observed G, derived as Rn-LE-H (Table S1)."*

(Line 327-328)

**Results:**

This section is well organized, followed by the order of the validation of modified MEP method at stations, comparisons of Bowen ratio formulas, evaluation of radiation and heat storage for model input, extended to global scale and analysis of new global estimates. However, additional analysis of spatial pattern variability across two different periods (before and after year 2003) can be considered.

**Response:** We gratefully appreciate for your valuable comment. We have analyzed the ET spatial pattern variability at two different periods (1988-2003, and 2003-2017). Based on your comments, we have revised Section 4.5. We have added two paragraphs to show the spatial variability before and after year 2003 (shown in Figure 11). The detailed revisions refer to the response to comment (11) below. For specific details regarding these revisions, please refer to our response to comment (11) below.

(10) Line 381, "This decision…", revised as "This choice…"

**Response:** Thank you for pointing this out. We have revised it as "This selection", and the revised text reads as follows.

*"This selection was based on the site's long-term observational records…"*

(Line 401)

(11) Line 620 - Line 632, From Fig.12, it seems that ocean ET from most ET datasets increased from 1988-2003, followed by fluctuations during 2003-2010, and then a consistent downward trend from 2010 to 2017. I am interested in the spatial variability for the periods 1988-2003 and 2003-2017, as 2003 appears to be a turning point for

evaporation changes. Providing a global spatial plot of ET trends for these two periods could be valuable for detecting spatial variability.

**Response:** We gratefully appreciate for your valuable comment. We have analyzed the ET spatial pattern variability at two different periods (1988-2003, and 2003-2017). The slowdown and transition of evaporation increase during 2003-2010 were consistent with the hiatus in global available energy and sea surface temperature. This phenomenon aligns with the concept of a "global warming hiatus" (Medhaug et al., 2017; Sung et al., 2023). This study supports the hypothesis that the hiatus in radiative forcing strongly affected the interannual variability of evaporation and surface temperature.

Based on your comments, we have made a major revision in Section 4.5. An additional analysis was conducted (before and after year 2003) and we have added more results to show the spatial variability in the Figure 11. Furthermore, we have added two paragraphs of content to analyze the spatial pattern variability at two different periods. The corresponding revisions in the Section 4.5 are listed as follows:

"However, global ocean evaporation experienced a notable shift around 2003, as illustrated in Figs.11b and 11c. The downward trend observed from 2003 to 2017 counteracted a significant portion of the growth trend that occurred during the previous 16 years (1988-2003), particularly evident in the mid-latitude regions (15ºS-20ºN). In the middle-to-low latitudes (0º-30ºN), nearly all ocean grids exhibited opposite trends around 2003. Spatially, regions that displayed the largest increasing trends during 1988-2003 transitioned to show the most substantial decreasing trends between 2003 and 2017. This includes regions associated with western boundary current systems, convergence zones of the East Australian Current and the South Equatorial Current, as well as equatorial regions of the Pacific and Atlantic Oceans (Fig. 11c). To further investigate the shift in ocean evaporation after 2003, we analyzed the interannual variability of global annual mean area-weighted evaporation using all available datasets (as shown in Fig.12)."

(Line 641-650)

"While different datasets revealed varying magnitudes of evaporation changes, most exhibited a similar temporal pattern: an increasing trend from 1988 to around 2003, followed by a hiatus during 2003-2010, and ultimately a decreasing trend after 2010 (Fig. 12a). Specifically, MEP indicated an increasing trend in evaporation of 3.58 mm/year from 1988 to 2010, followed by a decrease of 2.18 mm/year after 2010 (Fig. 12a). The slowdown and transition of evaporation during 2003-2010 aligned with the concept of a "global warming hiatus" (Medhaug et al., 2017; Sung et al., 2023), referring to the period when global mean surface air temperatures did not continue to rise between 1988 and 2012. Previous studies have proposed four potential explanations for this global warming hiatus: internal variability, external drivers, the Earth's response to $CO_2$, and radiative forcing (Medhaug et al., 2017). This study indicates that changes in radiative forcing (Fig. 12b) can significantly affect the interannual variability of evaporation (Fig. 12a) and surface temperature (Fig. 12c). This finding is consistent with previous research that attributed more than 50% of the uncertainty in MEP-modeled fluxes to the radiation term (Huang et al., 2017). Although surface temperature began to increase after 2012, the decrease in available energy remained the primary driver behind the decline in evaporation."
(Line 653-666, in Section 4.5)

Once again, thank you very much for your valuable comments and suggestions. We hope that the revisions in the manuscript and our accompanying responses will be sufficient to make our manuscript suitable for publication in *ESSD*.

---

## Author Comment (AC2)

**Responses to reviewers – ESSD-2024-420**

**"Global ocean surface heat fluxes revisited: A new dataset from maximum entropy production framework with heat storage and Bowen ratio optimizations"**, by Yong Yang, Huaiwei Sun, Jingfeng Wang, Wenxin Zhang, Gang Zhao, Weiguang Wang, Lei Cheng, Lu Chen, Hui Qin, Zhanzhang Cai, submitted to *Earth System Science Data*.

Dear Editors and Reviewers,

Thank you for your letter and for the reviewers' comments concerning our manuscript **ESSD-2024-420**. We appreciate editors and reviewers very much for the positive and constructive comments and suggestions.

Those comments are all valuable and very helpful for revising and improving our paper, as well as providing the important guidance for our researches. We have addressed all the comments carefully and made the revisions which we hope will meet your approval.

Below are our point-by-point responses to the comments raised by the referee. Comments of reviewers are marked with the blue color, and the corresponding responses are in black (begins with bold words of **"Response"**). Corresponding changes in the text of the manuscript appear in red font.

**Responses to Reviewer #2**

**Reviewer #2: General Comments:**
* * *
This research presents a new approach to estimating ocean surface heat fluxes using an improved Maximum Entropy Production (MEP) framework. The integration of heat storage effects and empirical Bowen ratio formulas into the MEP model significantly enhances the accuracy of the original MEP theory. Overall, the manuscript is clearly written, and the methodology is well-structured. However, I have several comments below:

1. The manuscript does not sufficiently acknowledge the limitations of the proposed method. Including a brief section on limitations would provide a more balanced perspective.

2. There is inconsistent use of past and present tense throughout the manuscript, particularly in the methods and results sections. The distinction between past (for methods) and present (for results or general facts) should be thoroughly reviewed and corrected.

**Response:** We feel great thanks for your professional review work on our article. As you are concerned, there are several problems related with article's structure and the writing style that need to be addressed. According to your constructive suggestions, we have made extensive corrections to our previous draft, the detailed corrections are listed below.

**(1) Responses to general comment 1:** We sincerely appreciate your constructive and valuable comments. We agree with you that include a section on limitations can provide a more balanced perspective. Reviewer 1 also suggested that include a brief "Limitation" section is needed. According to your suggestions, we have supplemented a subsection

of "5.4 Limitations" in the "5. Discussion" section to discuss the limitations of our proposed methods and the results. In this subsection of "Limitations", we have clarified the limitations on three perspectives: (1) Heat Storage Determination, (2) Bowen Ratio Improvement, and (3) Driving Mechanisms of Evaporation Trends. We have discussed these limitations in detail in this supplemented subsection. The corresponding revisions are as follows:

"**5.4 Limitations**

[revised manuscript text omitted]

"This discrepancy was likely due to the…"

"a polynomial regression method was employed to…"

"It was indicated that regions with significant increases…"

"This consistency was in line with previous…"

**"Discussion" section:**

"The sensitivity analysis revealed the…"

"the heat storage ($G$) exhibited seasonal variations…"

"The analysis indicated that $R_n$ significantly influenced…"

"For instance, a 10% decrease in positive $G$ yielded a 7.4% increase…"

"This formula was applicable for surface temperatures…"

"The coefficients were based on a mean α value of 1.26…"

"The improved MEP model provided precise $LE$ estimates…"

"This study applied the improved MEP model to ocean surface…"

**Specific Comments:**

(1) Title: Consider replacing "optimizations" with a more appropriate term.

Response: We gratefully appreciate for your valuable suggestion. In this study, we used the empirical Bowen ratio formulas to modify the original MEP Bowen ratio, we think this correction is an "adjustment" to the Bowen ratio of original MEP method. Therefore, we replaced the "optimizations" with the term "adjustment", and the revised title is as:

"Global ocean surface heat fluxes derived from the maximum entropy production framework accounting for ocean heat storage and Bowen ratio adjustments"
(Lin 1-3, in the "Title")

(2) Abstract: Briefly explain what the Bowen ratio is, as it is a key concept in the manuscript.

Response: Thanks for your helpful comment. We have added the explanation on the "Bowen ratio" in the "Abstract" section, and the revision is as:

"This study derived global ocean heat fluxes using the MEP theory, incorporating the effects of heat storage and adjustments to the Bowen ratio (the ratio of sensible heat to latent heat)"
(Line 27-29, in the "Abstract" section)

(3) The abstract emphasizes the adjustments to the Bowen ratio but should also explain why accounting for heat storage is critical.

**Response:** Thanks for your helpful comment. We agree with this comment that the importance of accounting for heat storage should be explained in the Abstract. We have added the explanation on the importance of considering heat storage as follows:

"Given the substantial heat storage capacity of the deep ocean, which can create temporal mismatches between variations in heat fluxes and radiation, it is crucial to account for heat storage when estimating heat fluxes."
(Line 25-27, in the "Abstract" section)

(4) L30: Replace "improve" with "improves".

Response:  Thank you for pointing this out. The Reviewer 1 also suggested that "(4) Lin 29-31, The statement needs to be re-organized". Therefore, we have rewritten this sentence and revised it as:

"By incorporating the heat storage effect and adjusting the Bowen ratio within the MEP model, the accuracy of the estimated heat fluxes was significantly improved, achieving an $R^2$ of 0.99…"
(Line 32-33, in the "Abstract" section)

(5) L33-L34: Provide more information to justify why the specific values mentioned here are significant.

**Response:** Thank you for your comment. This study provides new estimates of global annual average latent heat (93 $W \cdot m^{-2}$) and sensible heat flux (12 $W \cdot m^{-2}$). Compared with the estimates by the original MEP model (in Table 5), i.e., latent heat flux of 52.7 $W \cdot m^{-2}$ and sensible heat flux of 25.1 $W \cdot m^{-2}$, the improved MEP model has effectively corrected the underestimation of LE and overestimation of sensible heat by the original model. Therefore, we have added the reason why we mentioned the specific values here, and the revision is as:

"The improved MEP method successfully addressed the underestimation of *LE* and the overestimation of sensible heat by the original model, providing new global estimates

of *LE* at 93 W·m$^{-2}$ and sensible heat at 12 W·m$^{-2}$ for the annual average from 1988 to 2017."

(Line 34-37, in the "Abstract")

(6) L42: Revise "the water cycle and climate change" to "water cycle, and climate change".

**Response:** Thank you for your helpful comment. We have revised this sentence according to your comment as:

"This dataset provides a new benchmark for the ocean surface energy budget and is expected to be a valuable resource for studies on global ocean warming, sea surface-atmosphere energy exchange, the water cycle, and climate change."

(Line 42-44)

(7) L111-L112: Add more background information to explain why investigating the impact of heat storage is important.

**Response:** We gratefully appreciate for your valuable suggestion. According to your comment, we have added detailed background descriptions to explain the influences of heat storage and why investigating the impact of heat storage is important for estimating heat fluxes. Furthermore, the necessity about why incorporating heat storage in the MEP model is also described in Section 2.2.2 as: "This study introduced two key hypotheses: (1) The substantial heat storage capacity of the ocean can exert a significant influence on seasonal latent and sensible fluxes, potentially introducing bias to the MEP equations…". The content of the added background information is as follows:

"These substantial errors in MEP-estimated oceanic fluxes may be attributed to the lack of consideration of heat storage effects. The significant impact of heat storage in deep ocean water can introduce substantial bias in estimating seasonal evaporation rates

when using the Penman combination-based method (McMahon et al., 2013; Bai & Wang et al., 2023). For instance, deep-water bodies typically store heat during the spring and release it in the fall, which can lead to overestimation of evaporation in the summer and underestimation in the fall if changes in heat storage are not accounted for (Zhao & Gao, 2019; Morton, 1994). Therefore, when estimating heat fluxes using the Bowen ratio (Bo, defined as the ratio of H to LE) energy budget-based method (including the MEP method), it is essential to incorporate heat storage effects to ensure accurate partitioning of available energy."

(Line 91-99, in "Introduction" section)

References:

[1] Morton, F. I.: Evaporation research—a critical review and its lessons for the environmental sciences. Critical Reviews in Environmental Science and Technology, 24(3), 237–280. https://doi.org/10.1080/10643389409388467, 1994.

[2] McMahon, T. A., Peel, M. C., Lowe, L., Srikanthan, R., & McVicar, T. R.: Estimating actual, potential, reference crop and pan evaporation using standard meteorological data: a pragmatic synthesis. Hydrology and Earth System Sciences, 17(4), 1331-1363, https://doi.org/10.5194/hess-17-1331-2013, 2013.

[3] Bai, P., & Wang, Y.: The importance of heat storage for estimating lake evaporation on different time scales: Insights from a large shallow subtropical lake. Water Resources Research, 59, e2023WR035123. https://doi.org/10.1029/2023WR035123, 2023.

(In the "References" section)

(8) L120: Add "and" before "Rnl".

**Response:** Thank you for your helpful comment. We have revised it as:

"where $R_n$, $R_{ns}$, and $R_{nl}$ are net radiation, net shortwave radiation…"

(Line 138, in Section 2.1)

**Response:** Thank you for your helpful comment. We have revised it as:

"where $R_n$, $R_{ns}$, and $R_{nl}$ are net radiation, net shortwave radiation (the difference of incoming radiation $R_s^{\downarrow}$ and reflected solar radiation $R_s^{\uparrow}$), and net longwave radiation (the difference of incoming longwave radiation $R_l^{\downarrow}$ and outgoing longwave radiation $R_l^{\uparrow}$), respectively; $H$ is sensible heat, $LE$ is latent heat…"

(Line 140, in the Section 2.1)

**Response:** Thank you for your helpful comment. The is a writing mistake in the original draft, it should be "Since the previous study overlooked the calculation of lateral heat flux transported by ocean currents…", thanks for correcting this error. We have rewritten this sentence as:

"Since the lateral heat transport by ocean currents is zero at the global scale (Wang et al., 2021), $G$ can be regarded as equivalent to the change in ocean heat content or heat storage at the global level."

(Line 149-151, in the Section 2.1)

**Response:** Thank you for your helpful comment. We have revised it as:

"The MEP model simulates ocean surface heat fluxes…"

(Line 155, in the Section 2.2.1)

(12) L145, L177, L194: Remove the unnecessary space before "where".

**Response:** Thank you for pointing this out. We have removed the space before "where" in L145, L177, and L194.

(13) L161: This information is redundant and consider removing it.

**Response:** Thank you for pointing this out. We have removed this sentence in Line 161 and the revision is as:

"According to the MEP theory, the net solar radiation…"
(Line 179, in the Section 2.2.1)

(14) Subsection 2.3: This section should be reorganized. It primarily describes methods rather than data, which could confuse readers.

**Response:** Thank you for your constructive comment. We agree that this helpful suggestion will improve the structure of the manuscript. We have revised the relevant sentences and moved Subsection 2.3 to Section 3.1 in the "Data materials" section. The revised content now reads as follows:

"3.1 Input data for MEP model
The performance of both the original and improved Maximum Entropy Production (MEP) models is evaluated using observed data from in-situ buoy stations, as described in Section 3.2. The optimal empirical Bowen ratio formula for the MEP model is then determined through multi-site assessments. Subsequently, the improved MEP model is applied to estimate global heat fluxes using long-term remote sensing data, as detailed in Sections 3.3 and 3.4. Specifically, the input variables of net radiation, heat storage, and sea surface temperature driving the improved MEP model are derived from the J-OFURO3 dataset, spanning 1988 to 2017 with a spatial resolution of 0.25°, as outlined

in Section 4.3."

(Line 246-253, in Section 3.1)

**Response:** Thanks for your constructive comment. This paragraph aims to elaborate that we attempt to derive a heat storage dataset with the best accuracy by assessing the performances of individual datasets and fused dataset (AA and BTCH). According to your helpful comment, we have rewritten this paragraph to improve the clarity, making it more logical and coherent, and the revision is as:

"To drive the improved MEP model with high-quality input data, this study aims to obtain a heat storage dataset with optimal accuracy. The accuracy of the heat storage dataset was assessed using three approaches: (1) individual dataset, (2) a fused dataset generated using the Bayesian Three-Cornered Hat (BTCH) method (He et al., 2020), and (3) an ensemble means obtained through the arithmetic average (AA) method. Previous studies have demonstrated that the BTCH method effectively quantifies uncertainties across diverse datasets and improves accuracy by integrating multiple datasets without requiring prior knowledge (Long et al., 2017; Liu et al., 2021; Duan et al., 2024). A recent study further evaluated various data fusion methods, including BTCH and the AA method, for addressing uncertainties in global evapotranspiration estimates derived from different datasets. The findings revealed that while both BTCH and AA are effective in identifying lower-quality ET datasets, their ability to consistently produce higher-accuracy datasets remains uncertain and, in some cases, may even degrade the overall accuracy (Shao et al., 2022). The performance of these fusion methods is highly sensitive to the selection of input datasets. For instance, the AA method is particularly susceptible to the influence of lower-quality datasets, especially when the sample size is small. Similarly, the performance of BTCH diminishes as the error covariance among the included datasets increases."

(Line 228-241, in Section 2.4)

(16) L228: Better to include more elaboration of "For instance, the AA method…".

**Response:** Thanks for your helpful comment. We have elaborated on this viewpoint in detail, and described the advantages and limitations of the BTCH and AA methods in fusing datasets. The revision is as:

"The findings revealed that while both BTCH and AA are effective in identifying lower-quality ET datasets, their ability to consistently produce higher-accuracy datasets remains uncertain and, in some cases, may even degrade the overall accuracy (Shao et al., 2022). The performance of these fusion methods is highly sensitive to the selection of input datasets. For instance, the AA method is particularly susceptible to the influence of lower-quality datasets, especially when the sample size is small. Similarly, the performance of BTCH diminishes as the error covariance among the included datasets increases. Consequently, following a comparative analysis of the accuracy of individual, BTCH, and AA fusion datasets, this study selected the optimal heat storage dataset to drive the MEP model."
(Line 236-243, in Section 2.4)

(17) L326: Replace "exceeds" with "exceeding".

**Response:** Thank you for your helpful comment. We have revised it as:

"The original MEP model (without considering heat storage) showed a significant negative correlation between $LE$ and $H$ (with $R^2$ exceeding 0.65 as…)"
(Line 343-344, in Section 4.1)

(18) L331, L339-L341, L359-L360: These sentences should be rewritten to enhance clarity.

**Response:** Thanks for your helpful comments. We have rewritten these sentences to enhances clarity, ensures logical flow, and refines the language, the revisions are as:

"After incorporating the influence of heat storage effects (represented as *MEP_M*, as depicted in Fig.1b and Fig.1h), the MEP-simulated *LE* showed a good consistency with buoy observations, with an $R^2$ value of 0.97 and a reduced RMSE of 27 W·m$^{-2}$. However, the *MEP_M* method revealed a significant bias in the partitioning of *LE* and *H* from the available energy. Specifically, *LE* was underestimated by 25% (regression slope = 0.75), while H was overestimated by 46% compared to observations."
(Line 346-350, in Section 4.1)

"After incorporating the effects of heat storage, four variants of the MEP model were developed by replacing $B_{o*}$ with $B_{oa}$ derived from four different empirical formulas. These variants were defined as follows: *M_0.24* (where $B_{oa}=0.24 B_o^*$), *M_0.79* (where $B_{oa}=0.79 \ B_o^*-0.21$), *M_0.63* (where $B_{oa}=0.63B_o^*-0.15$), and *M_0.37* (where $B_{oa}=0.37B_o^*-0.05$). Adjusting the Bowen ratio significantly improved the accuracy of the energy flux estimates."
(Line 357-361, in Section 4.1)

"Specifically, the spatial patterns of simulated errors for the four variants of the MEP model were obtained (Fig. 2), along with the errors across different observational buoy arrays (Fig. 3). Overall, the four variants of the improved MEP models demonstrated relatively lower bias at low latitudes (10°S to 10°N) but exhibit larger bias in higher latitude regions (above 15°N), particularly at the KEO, WHOTS, and STRATUS buoy sites."
(Line 376-380, in Section 4.1)

(19) L381: Replace "decision" with "selection" or "choice".

**Response:** Thanks for your helpful comments. We have revised it as:

"This selection was based on the site's long-term…"
(Line 402, in Section 4.2)

(20) L385: Change "128 other sites" to "other 128 sites".

**Response:** Thanks for your helpful comments. We have revised it as:

"The improved MEP methods demonstrated comparable performance in estimating heat fluxes at the KEO site when compared with other 128 sites…"
(Line 404-405, in Section 4.2)

(21) L415: Change "closest to" to "were closest to".

**Response:** Thanks for your helpful comments. We have revised it as:

"Among the four empirical formulas, $M\_0.24$ simulated $LE$, $H$, and Bowen ratio values were closest to the observed values"
(Line 436-437, in Section 4.2)

(22) L427-L428: Replace "meters" with "m".

**Response:** Thanks for your helpful comments. We have revised it as:

"On deep ocean surfaces, with the most recent average depth estimate of 3,682m from NOAA satellite measurements, heat storage variations can influence depths up to 6,000m"
(Line 449-450, in Section 4.2)

(23) L437: Change "originates" to "originate" and add "the" before "heat storage".

**Response:** Thanks for your helpful comments. We have revised the tense to keep consistency throughout the paper as follows:

"Therefore, the prediction errors in *LE* and *H* originated from the inability to accurately quantify the heat storage."
(Line 458-459, in Section 4.2)

(24) L468: Replace "failed" with "fails".

**Response:** Thanks for your helpful comments. We have revised the tense to keep consistency throughout the paper as follows:

"The results revealed that while using the AA method (e.g., AA4) to fuse yields smaller errors compared to ERA5, MERRA2, and ΔOHC, it still failed to achieve the accuracy of the J-OFURO3 product"
(Line 490-491, in Section 4.3.2)

(25) L474: Capitalize "surface temperature" and italicize "SST".
**Responses:** Thank you for pointing this out. The revised text reads as follows.

"To ensure consistency with radiation data source, the Sea Surface Temperature (*SST*) data from J-OFURO3 was utilized for…"
(Line 497, in Section 4.3.2)

(26) Figure 8: Add p-values here as well.

**Response:** Thank you for pointing this out. According to your comment, we have added p-value ($p < 0.001$) in the Figure 8 as:

[Figure]

**Figure 8.** Scatter density plots of latent heat flux taken from different products versus observations from 129 buoy stations during the period 1988-2017: (a) Improved MEP model, (b) J-OFURO3, (c) ERA5, (d) MERRA2, and (e) OAFlux. A total of 15444 records of latent heat observations are included.

(In Figure 8)

(27) L611-L613: Rewrite this sentence for clarity.

**Response:** Thanks for your helpful comments. We have rewritten this sentence to improve the clarity, making it more logical and coherent. The sentence is revised as:

"The spatiotemporal patterns of evaporation were analyzed using the Sen's slope and Mann-Kendall test methods (Fig. 11). For the global ocean, approximately 74% of the regions showed an increasing trend, with about 27% of the grids exhibiting statistically significant increases ($p < 0.05$). In contrast, 26% displayed a decreasing trend, with only 5% of the grids showing statistically significant decrease ($p < 0.05$)."
(Line 630-634, in Section 4.5)

(28) Figure 11: Please confirm the trend legend colors used are consistent with the map colors.

**Response:** Thank you for pointing this out. The reviewer 1 also pointed out that the spatial-temporal variability of ocean evaporation should be analyzed at two separate periods. Therefore, we have revised the Figure 11 to include more result. Also, we have made sure that the trend legend colors used are consistent with the map colors. The revived Figure 11 is as:

[Figure]

**Figure 11.** Spatial distribution of multi-year trends in ocean evaporation estimated by the improved MEP method during (a) the period 1988-2017, (b) the period 1988-2003, (c) the period 2003-2017, and (d) the latitudinal average changes across three different periods.

(In Figure 11)

(29) L644: Add "the" before "impact".

**Responses:** Thank you for pointing this out. The revised text reads as follows.

"5.1 Quantifying the impact of heat storage and radiation with sensitivity analysis"

(Line 680, in Section 5.1)

**Responses:** Thank you for pointing this out. The revised text reads as follows.

"This study addresses the issue of underestimating *LE* by the original MEP model…"
(Line 746-747, in Section 5.3)

(31) L724-L726: This transition sentence needs revision for smoother flow.

**Response:** Thanks for your helpful comment. According to your comment, we have revised these sentences to improve the clarity, making it more logical and coherent. The revised text reads as follows:

"From a methodological perspective, the improved MEP method emerged as a novel approach for estimating energy fluxes that diverges from traditional bulk methods. The conventional bulk method requires multiple input parameters, including air temperature, specific humidity, wind speed, sea surface temperature, atmospheric pressure, and the observational height of all parameters (Fairall et al., 2003; Tomita et al., 2021). This method demands numerous input variables, and the estimated fluxes are highly sensitive to changes in temperature and humidity gradients. In contrast, the improved MEP model requires only net radiation, heat storage, surface temperature, and atmospheric pressure to simultaneously obtain latent and sensible heat fluxes, making it more flexible to operate and robust against variations in input variables."
(Line 752-759, in Section 5.3)

**Response:** Thanks for your helpful comment. The revised text reads as follows:
"Compared to the Penman model for water body evaporation (Tian et al., 2022; Zhao

et al., 2022; Bai et al., 2023), the major advantage of MEP method lies in…"

(Line 766-767, in Section 5.3)

Once again, thank you very much for your valuable comments and suggestions. We hope that the revisions in the manuscript and our accompanying responses will be sufficient to make our manuscript suitable for publication in *ESSD*.